# DiffPM: Diffusion-Based Generative Framework for Time Series Synthesis

## Abstract

Generative models for time series often fail to reconcile local accuracy with global structure. Autoregressive models accumulate errors over long horizons, while standard diffusion approaches can degrade long-range dependencies, resulting in samples with phase drift and weakened correlations. We introduce **DiffPM**, a non-autoregressive diffusion framework that resolves this tension by factorizing the generation process. DiffPM learns to model time series by explicitly separating them into low-frequency trends and high-frequency residuals, training two specialized, window-conditioned diffusion models. At inference, the models generate short, overlapping windows for each component in parallel. The individual segments are subsequently reassembled into a full sequence via a position-aware stitching mechanism that enforces inter-window consistency. This modular, decompose-and-recombine architecture allows specialized models to excel at local generation, while guaranteeing global coherence in the final synthesis. Our extensive evaluations demonstrate that DiffPM holds a performance advantage over existing methods as well as being considerably faster at inference. This advancement is quantified by marked improvements in metrics for distributional fidelity, such as Contextual FID, and enhanced temporal coherence over long-horizon benchmarks.

## 1 Introduction

Synthetic time-series data provides a powerful solution for augmenting limited datasets, validating analytical systems, and enabling privacy-preserving data sharing in sectors from finance to healthcare. A formidable barrier, however, prevents the full realization of this potential: the difficulty of generating samples that are both locally accurate and globally consistent. An effective generative model must replicate transient, short-term phenomena like signal spikes while simultaneously sustaining long-term structural properties such as seasonality, gradual trends, and multi-variate dependencies. This requirement to satisfy both local and global constraints exposes the inherent brittleness of many established generative techniques. In augmentation settings, practitioners require full-length, unconditional samples—not short imputations or forecast snippets. Most recent diffusion baselines are trained and evaluated on windowed tasks, making whole-series synthesis slow and operationally awkward.

The landscape of early deep generative techniques for sequential data was principally shaped by adversarial and variational paradigms. On one hand, GAN-based approaches, including TimeGAN (Yoon et al., 2019) and COT-GAN (Xu et al., 2020), sought to enforce temporal consistency through specialized loss functions, yet could not fully overcome the notorious challenges of mode collapse and unstable training dynamics. On the other hand, VAE-based models such as TimeVAE (Desai et al., 2021) offered a more reliable optimization process but frequently yielded reconstructions that lacked high-frequency detail due to oversmoothing. This fundamental tension between sample quality and training stability has led the field to explore diffusion models as a promising alternative. Despite their ability to generate high-fidelity samples without adversarial training, ensuring the integrity of long-range temporal dependencies in these models introduces a distinct set of research problems.

While powerful, applying diffusion models to time series reveals a core conflict: a single denoising network struggles to simultaneously learn low-frequency global structure and high-frequency local

details. This cross-scale interference can cause a model to sacrifice one for the other. Recent work has attempted to mitigate this in several ways. One line of research guides a single model with auxiliary losses or built-in decompositions, as in DiffusionTS (Yuan & Qiao, 2024) or multi-resolution schemes (Shen et al., 2024; Fan et al., 2024), but still risks interference within a monolithic architecture. Another approach offloads structural guidance to external data via retrieval (Liu et al., 2024), sacrificing the goal of a purely generative, self-contained model. A third path uses auto-regressive or block-wise dependencies (Hoogeboom et al., 2022; Arriola et al., 2025), trading full parallelism for sequential consistency. These are all partial solutions that highlight the need for a framework that is both fully generative and explicitly structured to avoid scale interference.

Beyond fidelity, **wall-clock feasibility** matters. When sampling entire sequences—particularly in **high-dimensional** datasets—window-wise or auto-regressive baselines require sequential passes and cross-window dependencies, leading to **substantially slower** generation for the same number of indices.

**DiffPM** tackles time–series generation by explicitly factorizing the process: we decompose each sequence into a low–frequency trend and a high–frequency residual, train two independent diffusion models specialized for these components, and at inference generate short overlapping windows for both in parallel before recombining them with position–aware stitching and smooth cross–fades. This decompose→generate→recombine pipeline avoids cross–scale interference that plagues single–network designs, delivering high local fidelity from the residual model while the trend model preserves long–range structure; within windows the process is fully non–autoregressive, and across windows it is embarrassingly parallel, enabling fast synthesis of long, globally coherent sequences without retrieval or extra supervision. Across diverse benchmarks, **DiffPM** improves distributional fit (e.g., contextual FID) and temporal coherence (cross–variable correlation) as well as sampling speed, establishing a strong new baseline for unconditional time–series generation.

Our contributions are as follows:

- We introduce **DiffPM**, a model that decomposes time series into trend and residual components and learns them with two specialized diffusion models. This explicitly mitigates the cross-scale interference common in monolithic architectures.

- We propose a window-based synthesis strategy where each local window is generated conditioned on its absolute position. This position-aware mechanism enables the model to achieve both high-fidelity local detail and long-range global coherence.

- Our framework is built for speed: all windows are generated independently and in parallel and stitched once, removing sequential bottlenecks and keeping whole-series synthesis feasible in high-dimensional settings.

## 2 RELATED WORK

The landscape of time-series generation has rapidly evolved from classical statistical models to sophisticated deep generative frameworks. Our work builds on this progression, but its core contribution—explicitly factorizing the generation process to resolve cross-scale interference—is best understood by tracing the challenges encountered by prior paradigms. We structure our review around this central theme.

### 2.1 EARLY DEEP GENERATIVE MODELS: THE EMERGENCE OF THE MULTI-SCALE PROBLEM

Early deep generative architectures for time series synthesis were largely dominated by two paradigms: Generative Adversarial Networks (GANs) and Variational Autoencoders (VAEs), both offering compelling data-driven alternatives to traditional methods. GAN-based frameworks, exemplified by TimeGAN (Yoon et al., 2019), leveraged a combination of adversarial and supervised learning signals within the latent space to enforce temporal consistency. Other works in this domain sought to better respect the underlying sequential structure through mechanisms like causal optimal transport (Xu et al., 2020). Concurrently, VAEs provided a more stable training framework. For example, TimeVAE (Desai et al., 2021) advanced the VAE approach by explicitly designing

the model with inductive biases for trend and seasonality, achieving robust performance even in low-data scenarios.

The successes of these foundational frameworks were consistently shadowed by a core methodological tension. On one hand, the adversarial training of GANs, while capable of generating high-fidelity details, is notoriously brittle and prone to issues like mode collapse. On the other hand, the stable optimization of the ELBO in VAEs frequently leads to an undesirable oversmoothing of high-frequency components, prioritizing global structural integrity at the expense of local precision. This persistent conflict between local detail generation and global structural modeling indicates the inherent difficulty for a singular, uniform model to capture the multi-scale temporal dynamics of time series data. Consequently, this observation strongly motivates the development of architectures designed to explicitly address disparate temporal scales.

## 2.2 DIFFUSION MODELS: A NEW PARADIGM AND A FAMILIAR CHALLENGE

Denoising diffusion models have recently emerged as the state of the art in high-fidelity synthesis for many domains, including audio waveforms (Kong et al., 2021) and images Rombach et al. (2022); Kasaei et al. (2025). Their application to time series was a natural next step. In contrast to GANs, they offer stable training and high-quality sample generation without adversarial objectives. At their core, these models operate non-autoregressively, learning to denoise an entire sequence in parallel at each step of a reverse process.

However, this parallel denoising process re-introduces the multi-scale challenge in a new form. A significant challenge arises when a single diffusion network is tasked with concurrently predicting noise across all frequency bands. This simultaneous prediction can lead to a phenomenon known as cross-scale interference, where the model's learning process is skewed. For instance, the optimization may favor the reconstruction of pronounced low-frequency trends over the preservation of subtle high-frequency information, or the inverse. Such an imbalance often manifests as artifacts in the generated signal, including phase discrepancies or a reduction in expected volatility. Recent work in time-series diffusion can be viewed as a collection of strategies to mitigate this very issue:

Many approaches retain a single diffusion network but attempt to guide its learning process. Diffusion-TS (Yuan & Qiao, 2024) established a strong baseline for unconditional generation by incorporating internal decomposition losses and frequency-domain penalties. Multi-resolution schemes denoise from coarse to fine scales (Shen et al., 2024; Fan et al., 2024), forcing the model to attend to different levels of granularity. While often effective, these methods still rely on a single set of network weights to manage all scales, risking residual interference.

Alternative strategies seek to impose external structure on the diffusion process. For instance, autoregressive diffusion models (ARDMs) (Hoogeboom et al., 2022) and their block-wise counterparts (Arriola et al., 2025) generate sequences one segment at a time. This design enforces sequential consistency at the cost of full parallelism. Another approach, retrieval-augmented diffusion (Liu et al., 2024), leverages examples from a reference database to steer the generation process, anchoring the output to relevant precedents.

## 2.3 POSITIONING.

Unlike methods that guide a *single* denoiser or rely on external retrieval, DiffPM *factorizes* generation into two independent diffusion models (trend/residual) and reassembles via position-conditioned, coverage-normalized overlap–add. This avoids cross–scale interference while remaining fully generative and non-autoregressive at the window level, with embarrassingly parallel sampling. See 4 for details.

## 3 PRELIMINARIES

We build on denoising diffusion probabilistic models (DDPMs) for unconditional generation (Ho et al., 2020). A fixed forward (noising) process gradually corrupts a clean sample $x_0 \in \mathbb{R}^{T \times D}$ over $S$ steps:

$$q(x_t \mid x_{t-1}) = \mathcal{N}\big(x_t;\ \sqrt{\alpha_t}\, x_{t-1},\ (1 - \alpha_t)\, \boldsymbol{I}\big), \qquad t = 1, \ldots, S,$$

where $\{\beta_t\}_{t=1}^S$ is a variance schedule with $\alpha_t = 1 - \beta_t$ and $\bar{\alpha}_t = \prod_{s=1}^t \alpha_s$. Marginalizing the chain yields a convenient closed form

$$q(x_t \mid x_0) = \mathcal{N}\big(x_t;\ \sqrt{\bar{\alpha}_t}\, x_0,\ (1 - \bar{\alpha}_t)\, \boldsymbol{I}\big),$$

so one can obtain a noisy $x_t$ directly via

$$x_t = \sqrt{\bar{\alpha}_t}\, x_0 + \sqrt{1 - \bar{\alpha}_t}\, \epsilon, \qquad \epsilon \sim \mathcal{N}(0, \boldsymbol{I}).$$

The reverse (denoising) process is parameterized by a neural network $\epsilon_\theta(x_t, t, c)$ that predicts the noise $\epsilon$ added at step $t$. The reverse transition is Gaussian,

$$p_\theta(x_{t-1} \mid x_t) = \mathcal{N}\big(x_{t-1};\ \mu_\theta(x_t, t, c),\ \sigma_t^2 \boldsymbol{I}\big),$$

with mean computed from the noise predictor

$$\mu_\theta(x_t, t, c) = \frac{1}{\sqrt{\alpha_t}} \left( x_t - \frac{1 - \alpha_t}{\sqrt{1 - \bar{\alpha}_t}}\, \epsilon_\theta(x_t, t, c) \right),$$

and variance $\sigma_t^2$ chosen per schedule (e.g., fixed, learned, or $\beta_t$). Training minimizes the (weighted) MSE between true and predicted noise

$$\mathcal{L}(\theta) = \mathbb{E}_{x_0, \epsilon, t} \Big[ w_t \big\| \epsilon - \epsilon_\theta(\sqrt{\bar{\alpha}_t}\, x_0 + \sqrt{1 - \bar{\alpha}_t}\, \epsilon,\ t,\ c) \big\|_2^2 \Big],$$

where, in theory, $t$ is drawn uniformly from $\{1, \dots, S\}$ and $w_t$ is a per-step weight (e.g., constant or cosine/SNR-aware) (Nichol & Dhariwal, 2021b). Sampling starts from $x_S \sim \mathcal{N}(0, \boldsymbol{I})$ and iteratively applies the reverse transitions $S \to S-1 \to \cdots \to 0$. Importantly, diffusion denoises *all* dimensions in parallel at each step, making it non-autoregressive in the spatial/temporal sense.

**Scope.** We adopt the standard DDPM setup above and specialize it in our method: a windowed, position-conditioned formulation with a two-branch factorization (trend/residual) and a stitching procedure for whole-series synthesis. Implementation details appear in Sec. 4; supporting analyses of schedules/SNR and stitching are deferred to Appendix. A and C.

## 4 METHOD

### 4.1 OVERVIEW

**DiffPM** is a non-autoregressive, *unconditional* diffusion framework that follows a *decompose* $\to$ *generate* $\to$ *recombine* pipeline. Given a multivariate series $\boldsymbol{x}_{1:T} \in \mathbb{R}^{T \times D}$, we split it into a low-frequency *trend* $\boldsymbol{\tau}_{1:T}$ and a high-frequency *residual* $\boldsymbol{r}_{1:T}$. We then train two *window-conditioned* diffusion models—one for $\boldsymbol{\tau}$, one for $\boldsymbol{r}$—on short, overlapping windows taken from each component. At inference, both models synthesize overlapping windows *in parallel* from Gaussian noise; windows are stitched by overlap-aware averaging to produce $\hat{\boldsymbol{\tau}}_{1:T^\star}$ and $\hat{\boldsymbol{r}}_{1:T^\star}$, and we form the final sample by

$$\hat{\boldsymbol{x}}_{1:T^\star} = \hat{\boldsymbol{\tau}}_{1:T^\star} + \hat{\boldsymbol{r}}_{1:T^\star}.$$

Training and sampling are configured per branch (trend/residual may use different cutoffs) to reduce gradient variance and align the learned score field with inference; see Appendix A for details on this branch-specific tuning. An architectural overview is shown in Fig. 1.

### 4.2 TREND–RESIDUAL DECOMPOSITION

Let $\boldsymbol{x}_{1:T} = (\boldsymbol{x}_1, \dots, \boldsymbol{x}_T)$ with $\boldsymbol{x}_t \in \mathbb{R}^D$. We compute a smooth trend by a centered moving average of width $K$ (odd, for clarity). For $t \in \{1, \dots, T\}$,

$$\boldsymbol{\tau}_t = \frac{1}{K} \sum_{j=t-(K-1)/2}^{t+(K-1)/2} \boldsymbol{x}_j, \tag{1}$$

implemented as a same-length discrete convolution with a uniform kernel (out-of-range indices treated as zeros). The residual is the high-frequency remainder

$$\boldsymbol{r}_t = \boldsymbol{x}_t - \boldsymbol{\tau}_t. \tag{2}$$

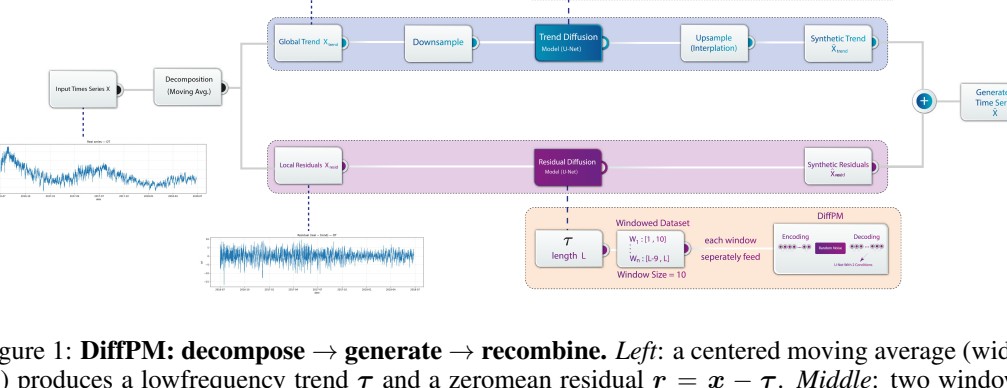
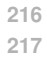

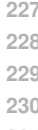

Figure 1: **DiffPM: decompose → generate → recombine.** *Left*: a centered moving average (width $K$) produces a lowfrequency trend $\boldsymbol{\tau}$ and a zeromean residual $\boldsymbol{r} = \boldsymbol{x} - \boldsymbol{\tau}$. *Middle*: two window-conditioned diffusion models (1D UNets with timestep and position embeddings) are trained independently on sliding windows of $\boldsymbol{\tau}$ and $\boldsymbol{r}$. *Right*: at inference, overlapping windows are sampled *in parallel* and stitched with overlapaware averaging, yielding $\hat{\boldsymbol{\tau}}, \hat{\boldsymbol{r}}$, and the final sample $\hat{\boldsymbol{x}} = \hat{\boldsymbol{\tau}} + \hat{\boldsymbol{r}}$.

Both sequences retain the original length and dimensionality. From this point onward, the *trend branch is trained on a downsampled trend* obtained by deterministic decimation (factor $K$); for notational simplicity we continue to denote this downsampled trend by $\boldsymbol{\tau}$.

This separation reduces cross-scale interference during learning: the trend model focuses on low-frequency, long-range structure while the residual model specializes in local variability. This deterministic analysis operator also underpins our likelihood: under this decomposition the training objective decomposes into a sum of two diffusion ELBOs (and admits an optional hierarchical residual); see Appendix B, Secs. B.3–B.4.

## 4.3 WINDOW-CONDITIONED DIFFUSION MODELS

We segment each series into fixed-length, overlapping windows. For length $T$, window size $L$, and stride $\Delta$ $(1 \leq \Delta < L)$, the start set is $\mathcal{S} = \{ s \in \{1, 1 + \Delta, 1 + 2\Delta, \ldots\} : s \leq T - L + 1\}$. From each $s \in \mathcal{S}$ we extract windows for the trend and residual: $\boldsymbol{\tau}_{s:s+L-1}, \boldsymbol{r}_{s:s+L-1} \in \mathbb{R}^{L \times D}$ (recall that the trend branch uses the *downsampled* trend; for notational simplicity we still write $\boldsymbol{\tau}$). Each window is conditioned on its absolute position via $c = (s, T)$, embedded and injected into the denoiser to preserve global context. We train on all windows independently and compute the loss per window *without* explicit overlap reweighting (see Fig. 2); this yields unbiased gradients for the per-window objective, preserves full parallelism, and matches our inference-time *coverage-normalized* stitching (4.4). Alternative overlap-aware formulations—such as inclusion–exclusion reweighting and conditional chaining—are derived in Appendix. B, Sec. B.6.

We adopt the DDPM objective on windows with position conditioning. Let $\boldsymbol{w} \in \mathbb{R}^{L \times D}$ denote a window (either trend or residual). The denoiser $\boldsymbol{\epsilon}_\theta(\boldsymbol{w}^{(t)}, t, c)$ predicts injected noise, and we minimize the simplified loss

$$\mathcal{L}(\theta) = \mathbb{E}_{\boldsymbol{w}^{(0)}, t, \boldsymbol{\epsilon}}\left[\left\|\boldsymbol{\epsilon} - \boldsymbol{\epsilon}_\theta(\boldsymbol{w}^{(t)}, t, c)\right\|_2^2\right], \quad (3)$$

with $t$ sampled uniformly each iteration (the forward/marginal forms follow 3). We train two independent models: $\theta_{\text{tr}}$ on trend windows and $\theta_{\text{res}}$ on residual windows. Hyperparameters may differ across branches; analyses of SNR/schedule choices appear in Appendix. A, while our default uses uniform $t$.

Both denoisers share a lightweight 1D UNet with residual blocks, GroupNorm, and SiLU activations. Convolutions run along time while treating the $D$ variables as channels; the initial projection and subsequent blocks thus mix cross-variable information naturally without attention. We form a

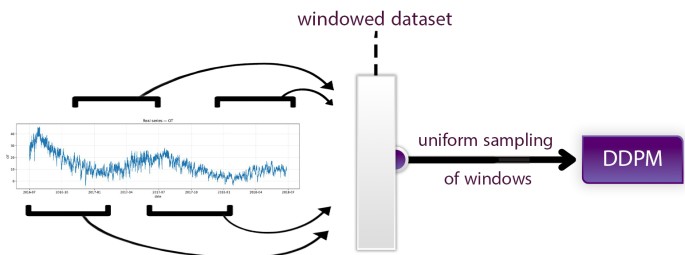

Figure 2: Overview of **DiffPM**: Break trend/residual into windows and train context-conditioned diffusion models; parallel window sampling with positional conditioning and smooth stitching.

conditioning vector $e_{\text{cond}} = e_t + e_{pos}$ by summing an MLP-transformed sinusoidal timestep embedding with a sinusoidal embedding of $c = (s, T)$, and inject it in each block via *additive* FiLM (bias only): for a normalized feature map $h$, we apply $h \leftarrow h + W\, e_{\text{cond}}$ followed by SiLU and a residual connection. This position-aware modulation is computationally cheap, preserves per-window non-autoregressiveness, and empirically stabilizes long-horizon structure by making absolute location available without altering the convolutional receptive field.

### 4.4 PARALLEL WINDOW GENERATION AND STITCHING

Our synthesis procedure begins by defining a lattice of overlapping windows. For a target length $T^\star$, window length $L$, and hop size $\Delta$, we use

$$\mathcal{S}^\star = \{\, s \in \{1, 1 + \Delta, 1 + 2\Delta, \ldots\} \; : \; s \le T^\star - L + 1 \,\}.$$

A key advantage of our framework is that all windows indexed by $\mathcal{S}^\star$ are sampled *independently*, enabling an embarrassingly parallel generation process. This design is consistent with our inference-time *coverage-normalized* stitching (defined below); alternative overlap-aware formulations (inclusion–exclusion reweighting, conditional chaining, inverse-variance blending) are presented in Appendix. B, Sec. B.6.

For each $s \in \mathcal{S}^\star$, we synthesize a local window for the trend and for the residual using their respective diffusion models. Concretely, starting from Gaussian noise and conditioning on $c = (s, T^\star)$, the trend branch generates

$$\hat{\boldsymbol{w}}^{(\text{tr})}_{s:s+L-1} \; \sim \; p_{\theta_{\text{tr}}}(\cdot \mid s, T^\star),$$

and the residual branch generates

$$\hat{\boldsymbol{w}}^{(\text{res})}_{s:s+L-1} \; \sim \; p_{\theta_{\text{res}}}(\cdot \mid s, T^\star).$$

We use the standard reverse diffusion chain (DDPM or DDIM) for each branch. The window-level synthesis is the superposition

$$\hat{\boldsymbol{z}}_{s:s+L-1} \; = \; \hat{\boldsymbol{w}}^{(\text{tr})}_{s:s+L-1} \; + \; \hat{\boldsymbol{w}}^{(\text{res})}_{s:s+L-1} \; \in \; \mathbb{R}^{L \times D}. \tag{4}$$

*Trend decimation/upsampling.* As the trend model is trained on a decimated trend (factor $K$), we generate trend windows on the decimated grid and then upsample the stitched trend back to length $T^\star$ (linear by default; sinc optional) before summation with the residual.

Once all windows $\{\hat{\boldsymbol{z}}_{s:s+L-1}\}_{s \in \mathcal{S}^\star}$ are generated, we assemble the final series $\hat{\boldsymbol{x}}_{1:T^\star}$ by *overlap–add with coverage normalization*. Let $\mathcal{W}(t) = \{\, s \in \mathcal{S}^\star : t \in [s, s + L - 1] \,\}$ be the windows covering absolute time index $t$. In our implementation, each contributing sample uses a *uniform* analysis weight, $w_{s,t} \equiv 1$, and we divide by the local coverage to avoid bias:

$$S^{(d)}_t = \sum_{s \in \mathcal{W}(t)} m^{(d)}_{s,t}\, \hat{z}^{(d)}_{s,t}, \qquad C^{(d)}_t = \sum_{s \in \mathcal{W}(t)} m^{(d)}_{s,t},$$

where $\hat{z}^{(d)}_{s,t}$ is the $d$-th channel of $\hat{\boldsymbol{z}}_{s:s+L-1}$ at absolute time $t$. We apply a deterministic validity mask using *global* bounds $(\ell, u)$ (per-channel bounds are possible but not used in our experiments):

$$m^{(d)}_{s,t} \; = \; \mathbf{1}\{\, \ell \le \hat{z}^{(d)}_{s,t} \le u \,\}.$$

The stitched output is

$$\hat{x}_t^{(d)} \;=\; \frac{S_t^{(d)}}{C_t^{(d)} + \varepsilon}, \qquad t = 1, \dots, T^\star, \;\; d = 1, \dots, D, \tag{5}$$

with a small $\varepsilon > 0$ for stability. Under strict COLA windows with partition-of-unity weights one obtains perfect reconstruction in the noiseless case; we instead use uniform weights with coverage normalization and rely on the *non-COLA* error bound in Appendix. C, which guarantees bounded reconstruction error, no worst-case error amplification, and variance reduction in overlaps—the exact regime we employ in all experiments.

Let $N_{\text{win}} = |\mathcal{S}^\star| \approx \lceil (T^\star - L)/\Delta \rceil + 1$ be the number of windows and $S$ the number of reverse steps. The total work scales as $\mathcal{O}(N_{\text{win}} S)$, but wall-clock is dominated by the cost of *one* window's reverse chain thanks to parallel sampling; stitching is linear-time in $T^\star$.

## 5 RESULTS AND EXPERIMENTS

We evaluate **DiffPM** as an *unconditional* multivariate time–series generator. Our experimental program measures (i) how well synthetic samples match the real data distribution, (ii) whether temporal and cross–variable dependencies are preserved, and (iii) whether synthetic data are useful for downstream modeling via train–synthetic–test–real (TSTR). We compare to strong baselines spanning GAN-, VAE-, and diffusion-based families: TimeGAN, TimeVAE, Diffwave, DiffTime, Diffusion-TS, and Cot-GAN.

### 5.1 DATASETS

We use four benchmarks covering finance, energy systems, and neuroimaging simulations:

**Stocks.** Daily Google stock prices from 2004–2019, one record per day with *6* features (open, high, low, close, adjusted close, volume). **ETTh.** Transformer-temperature/loads recorded every 15 minutes between July 2016 and July 2018; standard multivariate split used in prior work. **Energy.** UCI appliances energy dataset with *28* channels (indoor/outdoor conditions and appliance loads). **fMRI.** Realistic simulations of BOLD time series for causal discovery; we use a standard configuration with *50* variables.

All methods are trained on the training split only and evaluated on held-out test sets; we report mean±std over multiple seeds. Windowing $(L, \Delta)$ and stitching follow our implementation: fixed-length windows with position conditioning and *coverage-normalized* overlap–add using uniform weights (see 4.3, 4.4). Appendix C provides the non-COLA error bound we rely on; Appendix B discusses alternative overlap-aware formulations that we do not use in these experiments.

### 5.2 EVALUATION METRICS

We adopt four standard criteria that together probe distributional fidelity, dependency structure, and downstream usefulness:

**Predictive score (TSTR)** (Yoon et al., 2019). A next–step predictor is trained *on synthetic data* and evaluated *on real* test sequences; we report the prediction error (lower is better), isolating the utility of synthetic data for model training.

**Context–FID (cFID)** (Paul et al., 2022). A Fréchet distance computed on contextual representations of subsequences; assesses local–in–context realism and diversity (lower is better).

**Correlational score** (Ni et al., 2020). Absolute Frobenius error between cross–correlation matrices computed from real vs. synthetic sequences; probes temporal and cross–variable dependency preservation (lower is better).

For overlap consistency and local realism, we rely on the stitching analysis for our *coverage-normalized* uniform-weight overlap–add: Appendix C establishes a non-COLA error bound (bounded reconstruction error, no worst-case amplification, and variance reduction in overlaps), which supports the use of contextual metrics. For completeness, the same appendix shows perfect reconstruction under strict COLA windows—conditions we do not enforce in our experiments.

Table 1: **Quantitative comparison on four datasets.** Lower is better. Best in **bold**, second best underlined.

| Metric | Methods | Stock | ETTh | Energy | fMRI |
|---|---|---|---|---|---|
| Context-FID Score (lower is better) | **DiffPM (ours)** | **0.097 ± 0.012** | 0.210 ± 0.010 | 0.142 ± 0.005 | **0.095 ± 0.007** |
| | Diffusion-TS | 0.147 ± 0.025 | **0.116 ± 0.010** | **0.089 ± 0.024** | 0.105 ± 0.006 |
| | TimeGAN | 0.103 ± 0.013 | 0.300 ± 0.013 | 0.767 ± 0.103 | 1.292 ± 0.218 |
| | TimeVAE | 0.215 ± 0.035 | 0.805 ± 0.186 | 1.631 ± 1.142 | 14.449 ± 0.969 |
| | Diffwave | 0.232 ± 0.032 | 0.873 ± 0.061 | 1.013 ± 1.131 | 0.244 ± 0.018 |
| | DiffTime | 0.236 ± 0.074 | 0.299 ± 0.044 | 0.279 ± 0.045 | 0.340 ± 0.015 |
| | Cot-GAN | 0.408 ± 0.086 | 0.980 ± 0.071 | 1.039 ± 0.028 | 7.813 ± 0.550 |
| Correlational Score (lower is better) | **DiffPM (ours)** | 0.010 ± 0.001 | **0.032 ± 0.001** | **0.361 ± 0.049** | **0.331 ± 0.008** |
| | Diffusion-TS | **0.004 ± 0.001** | 0.049 ± 0.008 | 0.856 ± 0.147 | 1.411 ± 0.042 |
| | TimeGAN | 0.063 ± 0.005 | 0.201 ± 0.020 | 4.010 ± 1.104 | 23.502 ± 0.039 |
| | TimeVAE | 0.095 ± 0.008 | 0.111 ± 0.020 | 1.688 ± 2.226 | 17.296 ± 2.526 |
| | Diffwave | 0.036 ± 0.002 | 0.175 ± 0.006 | 5.001 ± 1.154 | 3.927 ± 0.349 |
| | DiffTime | 0.006 ± 0.002 | 0.067 ± 0.015 | 1.158 ± 0.095 | 1.507 ± 0.051 |
| | Cot-GAN | 0.087 ± 0.004 | 0.249 ± 0.009 | 3.164 ± 0.061 | 26.824 ± 0.449 |
| Predictive Score (TSTR) (lower is better) | **DiffPM (ours)** | **0.022 ± 0.001** | **0.107 ± 0.001** | 0.261 ± 0.031 | 0.102 ± 0.000 |
| | Diffusion-TS | 0.036 ± 0.001 | 0.119 ± 0.002 | **0.250 ± 0.000** | **0.099 ± 0.000** |
| | TimeGAN | 0.039 ± 0.000 | 0.126 ± 0.004 | 0.273 ± 0.002 | 0.126 ± 0.002 |
| | TimeVAE | 0.039 ± 0.000 | 0.124 ± 0.004 | 0.302 ± 0.002 | 0.113 ± 0.002 |
| | Diffwave | 0.039 ± 0.000 | 0.123 ± 0.004 | 0.297 ± 0.002 | 0.108 ± 0.002 |
| | DiffTime | 0.038 ± 0.001 | 0.121 ± 0.004 | 0.274 ± 0.002 | 0.106 ± 0.001 |
| | Cot-GAN | 0.047 ± 0.001 | 0.129 ± 0.007 | 0.259 ± 0.000 | 0.185 ± 0.003 |

## 5.3 BASELINES AND PROTOCOL

We include representative models across families: TimeGAN (adversarial+supervised), TimeVAE (variational), Diffwave/DiffTime/Diffusion-TS (diffusion), and Cot-GAN (optimal-transport–guided GAN). For each dataset, all models use identical train/val/test splits and, where applicable, the same windowing configuration (length $L$, stride $\Delta$). For diffusion-based methods we standardize stitching to the *coverage-normalized* uniform overlap–add used by DiffPM (4.4), so differences are not attributable to stitching.

Hyperparameters follow each method's recommended settings; early stopping uses validation metrics. Unless stated otherwise, diffusion timesteps are sampled *uniformly* and the cosine schedule is used (no banded time sampling in main results); analyses of variance and banded sampling are provided in Appendix A.

For TSTR, the downstream predictor architecture is fixed across methods to attribute differences to data quality. Each experiment is repeated with multiple random seeds; we report mean±std.

Augmentation ablations (Appendix. D) use *identical* splits, fixed downstream models, and scalers fit on *real-train* only; causal preprocessing is enforced (no look-ahead). DiffPM synthetics are stitched with the same *coverage-normalized* overlap–add and lightly post-processed to match train statistics (rolling mean smoothing, per-feature rescale; no clipping unless stated). *Only* the presence of DiffPM synthetics in the training split differs.

Generation-time benchmarks for DiffPM and baselines are in Appendix. E: we report wall-clock per generated index and throughput scaling with length and dimensionality under identical windowing $(L, \Delta)$ and each method's standard sampler/step settings.

## 5.4 MAIN QUANTITATIVE RESULTS

All DiffPM results use the *coverage-normalized* uniform overlap–add stitching described in 4.4; Appendix C establishes the non-COLA error bound that supports this choice. Our training objective is the standard simplified DDPM loss applied *independently* to trend and residual windows (Eq. 3); Appendix B, Secs. B.3–B.4, derives the two-branch windowed ELBO and discusses a hierarchical variant, which we do not employ in our experiments. The primary quantitative results against established baselines across four datasets are shown in Table 1.

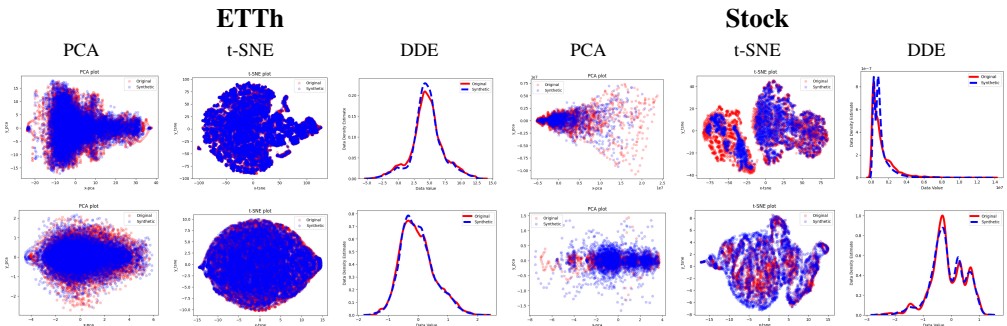

Figure 3: **Qualitative comparison on ETTh and Stock.** Top row: Diffusion-TS. Bottom row: DiffPM. Closer real–synthetic overlap in PCA/t-SNE and cleaner DDE geometry indicate higher distributional fidelity and temporal coherence.

## 5.5 QUALITATIVE VISUALIZATIONS

Following Yuan & Qiao (2024), we compare real and synthetic distributions via low-dimensional embeddings and a dynamical delay-embedding (DDE) view. For each dataset (**ETTh**, **Stock**), we present side-by-side overlays for **PCA** and **t-SNE** (real vs. model samples), and a **DDE** plot emphasizing temporal structure (*see* Fig. 3). Greater real/synthetic overlap indicates stronger distributional fidelity and coverage, while compact DDE geometry without spurious modes reflects preserved long-range dynamics. Qualitative seam inspection is consistent with our *coverage-normalized* stitching analysis.

**Reproducibility Statement.** We release an anonymized code archive with training/inference scripts, exact hyperparameters, and seeds to reproduce all tables and figures (supplementary materials). The method is fully specified in 4 (decomposition, windowing, conditioning, stitching), with objective and schedule choices in 4.3 and 4.4. Theoretical clarifications appear in Appendix. B (two-branch/windowed ELBO), Appendix. A (SNR view and time-sampling analysis), and App. C (stitching guarantees and non-COLA bound). Datasets and preprocessing (window sizes/strides, trend decimation factor, standardization on *real-train* only) are detailed in 5.1; evaluation metrics and protocols are in 5.2 and 5.3. Additional augmentation ablations and scripts are provided in Appendix. D.

## 6 CONCLUSION

We introduced **DiffPM**, a factorized, window-conditioned diffusion framework that learns low-frequency trend and high-frequency residual with two independent denoisers and stitches parallel window samples via coverage-normalized overlap–add. This decompose→generate→recombine design mitigates cross-scale interference, preserves long-range structure, and remains practical for whole-series synthesis in high-dimensional settings. Across four benchmarks, DiffPM delivers strong distributional fidelity and dependency preservation; augmentation ablations further show downstream gains (Appendix. D). Future work includes hierarchical residual conditioning, banded time sampling during training, and learned analysis windows for stitching.

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

# A  VARIANCE ANALYSIS AND NOISE–SCHEDULE TUNING FOR DIFFPM

This appendix formalizes the rationale behind two key strategies for improving the training stability of DIFFPM: sampling diffusion times from a restricted "middle band" and reshaping the noise schedule. Our analysis centers on the standard simplified DDPM loss, where we make explicit the sources of stochasticity and their relationship to the noise schedule and time-sampling distribution. We begin by establishing the setting and notation, then proceed to derive precise variance decompositions. This framework allows us to demonstrate how banded time sampling reduces training variance and to introduce an unbiased alternative via importance sampling.

Each branch of DIFFPM (trend and residual) is treated as an independent DDPM with $S$ discrete diffusion steps. For a data window $\boldsymbol{x}^{(0)} \in \mathbb{R}^{L \times D}$ and a timestep $t \in \{0, \ldots, S-1\}$, the forward noising process is defined by the conditional distribution:

$$q(\boldsymbol{x}^{(t)} \mid \boldsymbol{x}^{(0)}) = \mathcal{N}\big(\boldsymbol{x}^{(t)}; \sqrt{\bar{\alpha}_t}\,\boldsymbol{x}^{(0)}, (1 - \bar{\alpha}_t)\,\boldsymbol{I}\big), \qquad \bar{\alpha}_t = \prod_{s=0}^{t} \alpha_s, \ \ \alpha_s = 1 - \beta_s. \tag{6}$$

Here, $\boldsymbol{I}$ is the identity matrix in $\mathbb{R}^{(LD) \times (LD)}$ (assuming the window is flattened), and $\boldsymbol{\epsilon} \sim \mathcal{N}(\boldsymbol{0}, \boldsymbol{I})$ is a standard Gaussian noise vector of matching shape. All subsequent $\|\cdot\|_2$ norms operate on this flattened $LD$-vector representation. The training objective is based on the simplified noise-prediction loss:

$$\boldsymbol{x}^{(t)} = \sqrt{\bar{\alpha}_t}\,\boldsymbol{x}^{(0)} + \sqrt{1 - \bar{\alpha}_t}\,\boldsymbol{\epsilon}, \quad \boldsymbol{\epsilon} \sim \mathcal{N}(\boldsymbol{0}, \boldsymbol{I}), \qquad \ell(\theta; \boldsymbol{x}^{(0)}, t, \boldsymbol{\epsilon}) = \big\|\boldsymbol{\epsilon} - \boldsymbol{\epsilon}_\theta(\boldsymbol{x}^{(t)}, t)\big\|_2^2, \tag{7}$$

which is optimized over the full expectation:

$$L(\theta) = \mathbb{E}_{t \sim p(t)}\, \mathbb{E}_{\boldsymbol{x}^{(0)} \sim \mathcal{D},\, \boldsymbol{\epsilon} \sim \mathcal{N}(\boldsymbol{0}, \boldsymbol{I})}\big[\ell(\theta; \boldsymbol{x}^{(0)}, t, \boldsymbol{\epsilon})\big]. \tag{8}$$

Unless otherwise specified, $p(t)$ is the discrete uniform distribution over $\{0, \ldots, S-1\}$, i.e., $p(t) = 1/S$. We define the per-timestep expected loss as $L_t(\theta) := \mathbb{E}_{\boldsymbol{x}^{(0)}, \boldsymbol{\epsilon}}[\ell(\theta; \boldsymbol{x}^{(0)}, t, \boldsymbol{\epsilon})]$ and the per-timestep mean gradient as $\boldsymbol{\mu}_t(\theta) := \mathbb{E}_{\boldsymbol{x}^{(0)}, \boldsymbol{\epsilon}}[\nabla_\theta \ell(\theta; \boldsymbol{x}^{(0)}, t, \boldsymbol{\epsilon})]$. All branches of our model employ the cosine noise schedule introduced by Nichol & Dhariwal (2021a):

$$\bar{\alpha}_t = \frac{\cos^2\big(\frac{t/S+s}{1+s} \cdot \frac{\pi}{2}\big)}{\cos^2\big(\frac{s}{1+s} \cdot \frac{\pi}{2}\big)}, \qquad \beta_t = \min\big(1 - \bar{\alpha}_t/\bar{\alpha}_{t-1}, 0.999\big), \ \ s > 0. \tag{9}$$

We set $\bar{\alpha}_{-1} = 1$ for consistency and compute $\beta_t$ for $t \geq 1$ with clipping at $0.999$. We use the standard offset $s = 0.008$.

## A.1  SOURCES OF VARIANCE IN STOCHASTIC GRADIENT ESTIMATION

The training process relies on a Monte Carlo estimator for the gradient of the objective $L(\theta)$. For a minibatch of size $B$, this estimator is given by:

$$\widehat{\boldsymbol{g}}_B(\theta) := \frac{1}{B} \sum_{i=1}^{B} \boldsymbol{g}\big(\theta; \boldsymbol{x}_i^{(0)}, t_i, \boldsymbol{\epsilon}_i\big), \qquad t_i \overset{\text{i.i.d.}}{\sim} p(t), \ \ \boldsymbol{x}_i^{(0)} \sim \mathcal{D}, \ \ \boldsymbol{\epsilon}_i \sim \mathcal{N}(\boldsymbol{0}, \boldsymbol{I}), \tag{10}$$

where $\boldsymbol{g}(\theta; \boldsymbol{x}^{(0)}, t, \boldsymbol{\epsilon}) := \nabla_\theta \ell(\theta; \boldsymbol{x}^{(0)}, t, \boldsymbol{\epsilon})$ is the per-sample, per-timestep gradient. Assuming standard regularity conditions that permit interchanging differentiation and expectation, this estimator is unbiased, as shown by the following chain of expectations:

$$\mathbb{E}\big[\widehat{\boldsymbol{g}}_B(\theta)\big] = \mathbb{E}_{t, x^{(0)}, \boldsymbol{\epsilon}}\big[\boldsymbol{g}(\theta; \boldsymbol{x}^{(0)}, t, \boldsymbol{\epsilon})\big] = \mathbb{E}_t\big[\boldsymbol{\mu}_t(\theta)\big] = \nabla_\theta L(\theta).$$

The covariance of this minibatch estimator reveals two distinct sources of variance. By the law of total covariance and assuming independence across minibatch samples, we have:

$$\begin{aligned}
\text{Cov}\big(\widehat{\boldsymbol{g}}_B(\theta)\big) &= \frac{1}{B} \text{Cov}_{t, \boldsymbol{x}^{(0)}, \boldsymbol{\epsilon}}\big(\boldsymbol{g}(\theta; \boldsymbol{x}^{(0)}, t, \boldsymbol{\epsilon})\big) \\
&= \frac{1}{B}\Big( \underbrace{\mathbb{E}_t\big[\text{Cov}_{\boldsymbol{x}^{(0)}, \boldsymbol{\epsilon}}(\boldsymbol{g} \mid t)\big]}_{\text{Within-timestep variance}} + \underbrace{\text{Cov}_t\big(\boldsymbol{\mu}_t(\theta)\big)}_{\text{Time-sampling variance}} \Big).
\end{aligned} \tag{11}$$

The first term reflects variance from data sampling and noise injection at a fixed timestep. The second term, however, arises from the variability of the mean gradient $\boldsymbol{\mu}_t$ across different timesteps. [1] Using the identity $\mathbb{E}\|X - \mathbb{E}X\|_2^2 = \mathrm{Tr}(\mathrm{Cov}(X))$, we can express this matrix decomposition as a scalar variance decomposition:

$$\mathbb{E}\big[\|\widehat{\boldsymbol{g}}_B - \mathbb{E}\widehat{\boldsymbol{g}}_B\|_2^2\big] = \frac{1}{B}\Big(\mathbb{E}_t\,\mathbb{E}_{\boldsymbol{x}^{(0)},\boldsymbol{\epsilon}}\big[\|\boldsymbol{g} - \boldsymbol{\mu}_t\|_2^2 \mid t\big] \; + \; \mathbb{E}\big[\|\boldsymbol{\mu}_t - \mathbb{E}_t\boldsymbol{\mu}_t\|_2^2\big]\Big). \tag{12}$$

This decomposition highlights a critical issue: the total variance is lower-bounded by the time-sampling term, which does not vanish as $B \to \infty$. If $\boldsymbol{\mu}_t$ varies substantially with $t$, this second term can dominate and destabilize training.

To formalize the connection between the loss spread and gradient spread, we can bound the norm of the mean gradient. Let the denoising model be $f_\theta(\boldsymbol{x}^{(t)}, t) := \boldsymbol{\epsilon}_\theta(\boldsymbol{x}^{(t)}, t)$ and its parameter-Jacobian be $J_\theta$. Assuming its operator norm is uniformly bounded, $\|J_\theta(\boldsymbol{x}^{(t)}, t)\|_{\mathrm{op}} \leq M < \infty$, we have:

$$\|\boldsymbol{\mu}_t(\theta)\| = \big\|2\,\mathbb{E}[J_\theta^\top (f_\theta - \boldsymbol{\epsilon})]\big\| \; \leq \; 2M\,\sqrt{L_t(\theta)}. \tag{13}$$

This inequality rigorously links the mean gradient magnitude to the per-timestep loss. To relate this to the variation across timesteps, we can introduce a fixed baseline $\overline{L}$ (e.g., $\overline{L} = \mathbb{E}_t[L_t]$) and apply the triangle inequality for square roots:

$$\|\boldsymbol{\mu}_t(\theta)\| \; \leq \; 2M\Big(\sqrt{|L_t(\theta) - \overline{L}|} \; + \; \sqrt{\overline{L}}\Big). \tag{14}$$

Together, these inequalities formalize the intuition that a wide spread in $L_t(\theta)$ across $t$ implies significant variation in $\boldsymbol{\mu}_t(\theta)$, thus inflating the time-sampling variance term.

## A.2 Banded Time Sampling and the Bias–Variance Tradeoff

Based on this analysis, we define banded time sampling. For $S$ diffusion steps, we fix a lower and upper bound $0 < \beta < \omega < 1$ and define the middle band $\mathcal{T}_{\beta,\omega} = \{\lfloor \beta S \rfloor, \ldots, \lfloor \omega S \rfloor\}$. We replace the uniform sampling distribution $p(t)$ with $p_{\beta,\omega}(t)$, the uniform law on this set. To quantify the effect, we define the trace of the covariance of the mean gradient over this band:

$$\mathcal{G}_{\beta,\omega}(\theta) := \mathrm{Tr}\,\mathrm{Cov}_{t\sim p_{\beta,\omega}}\big(\boldsymbol{\mu}_t(\theta)\big). \tag{15}$$

Over the feasible region of $(\beta, \omega)$, this function is piecewise-constant and minimizers exist. Combining the bound in equation 13 with Jensen's inequality provides a direct link between the average loss over the band and an upper bound on this gradient covariance:

$$\mathcal{G}_{\beta,\omega}(\theta) = \mathrm{Tr}\,\mathrm{Cov}_{t\sim p_{\beta,\omega}}\big(\boldsymbol{\mu}_t(\theta)\big) \; \leq \; \mathbb{E}_{t\sim p_{\beta,\omega}}\big[\|\boldsymbol{\mu}_t(\theta)\|^2\big] \; \leq \; 4M^2\,\mathbb{E}_{t\sim p_{\beta,\omega}}\big[L_t(\theta)\big]. \tag{16}$$

This result provides a powerful insight: to minimize an upper bound on the time-sampling component of the gradient variance, one should choose a band $(\beta, \omega)$ that minimizes the average loss over that band. Training directly on this band introduces a beneficial bias, focusing model capacity on the most informative timesteps, while simultaneously reducing stochastic gradient variance. To align inference, we start the reverse diffusion process at $t_0 = \lfloor \omega S \rfloor$.

## A.3 Unbiased Estimation with Importance Sampling

If optimizing the original objective $L(\theta)$ is strictly required, one can use importance sampling (IS). Let $\boldsymbol{g}_i$ be a shorthand for the per-sample gradient. The IS estimator is:

$$\widehat{\boldsymbol{g}}_B^{\mathrm{IS}}(\theta) \; := \; \frac{1}{B}\sum_{i=1}^{B}\underbrace{\frac{p(t_i)}{p_{\beta,\omega}(t_i)}}_{w(t_i)}\,\boldsymbol{g}_i, \qquad t_i \sim p_{\beta,\omega}. \tag{17}$$

This estimator is unbiased, but at the cost of increased variance. Its covariance is given by:

$$\mathrm{Cov}(\widehat{\boldsymbol{g}}_B^{\mathrm{IS}}) = \frac{1}{B}\Big(\mathbb{E}_{t\sim p_{\beta,\omega}}\big[w(t)^2\,\mathbb{E}_{\boldsymbol{x}^{(0)},\boldsymbol{\epsilon}}[\boldsymbol{g}\boldsymbol{g}^\top \mid t]\big] - \big(\nabla_\theta L\big)\big(\nabla_\theta L\big)^\top\Big). \tag{18}$$

---

[1] If minibatches are constructed from overlapping windows from the same time series, the samples are not strictly i.i.d. While the estimator remains unbiased, the $1/B$ scaling becomes a heuristic upper bound due to non-zero cross-sample covariances.

The variance of the IS estimator scales with the second moment of the weighted gradient, $\mathbb{E}_{t \sim p_{\beta,\omega}}[w(t)^2 \|\boldsymbol{g}\|^2]$. In our case, the weight $w(t) = \frac{S}{|\mathcal{T}_{\beta,\omega}|} > 1$ is constant in the band. The increased variance of importance sampling thus necessitates practical safeguards. To control it, one should ensure the band is of non-trivial width to keep weights moderate, and it is advisable to pair IS with a larger batch size $B$. For these reasons, we favor the biased banded objective for its superior training stability.

### A.4 Practical Implications and Branch-Specific Tuning

The problematic nature of extreme timesteps can be understood from a signal-to-noise ratio perspective, where $\mathrm{SNR}(t) = \bar{\alpha}_t/(1 - \bar{\alpha}_t)$. For small $t$, the SNR is high, the denoising task is trivial, and gradients are small; batches with many such steps contribute to within-timestep variance without moving parameters meaningfully. For large $t$, the SNR is low, the input is nearly pure noise, and the loss becomes large but highly stochastic, causing large, erratic mean gradients that inflate time-sampling variance.

The cosine schedule is designed to mitigate these issues by smoothing the decay of $\bar{\alpha}_t$, but it does not eliminate the problematic tails. Banded sampling is an orthogonal variance reduction technique that complements the cosine schedule by explicitly removing these tails from the sampling distribution.

For the two-branch architecture of DIFFPM, this framework suggests branch-specific tuning. The trend and residual components often operate in different effective SNR regimes; residual windows are typically higher-frequency and lower-amplitude, meaning their signal degrades more quickly as noise is added. This observation leads to a practical heuristic: use a common lower cutoff $\beta$ for both branches, but an earlier upper cutoff for the residual branch ($\omega_{\mathrm{R}} \leq \omega_{\mathrm{T}}$). This branch-specific tuning, guided by a simple calibration, directly reduces the time-sampling variance from equation 11 and can improve convergence without altering the model architecture or loss function. In summary, by restricting time sampling to a "middle band," we strategically introduce a beneficial bias that reduces a dominant source of gradient variance, stabilizing and accelerating training.

## B Modified ELBO for Windowed Two-Branch DiffPM

This appendix provides a rigorous formulation of the evidence lower bound (ELBO) used by DiffPM when modeling a time series as the sum of two components—a smooth trend and a higher-frequency residual—and when training on fixed-length windows (with or without overlap). We derive the ELBO under (i) independent-component and (ii) hierarchical (conditional) generative assumptions. We then extend the analysis to windowed training, including overlapping windows via two alternatives: inclusion–exclusion reweighting and conditional (blockwise) chaining.

### B.1 Notation and Setup

Let an observed sequence window be $\mathbf{x}_0 \in \mathbb{R}^{L \times D}$ (length $L$, $D$ channels). We posit a deterministic decomposition

$$\mathbf{x}_0 = \boldsymbol{\tau}_0 + \mathbf{r}_0, \qquad \boldsymbol{\tau}_0 = \mathcal{T}(\mathbf{x}_0), \qquad \mathbf{r}_0 = \mathbf{x}_0 - \mathcal{T}(\mathbf{x}_0), \tag{19}$$

where $\mathcal{T} : \mathbb{R}^{L \times D} \to \mathbb{R}^{L \times D}$ is a fixed trend extractor (e.g., a centered moving average). DiffPM trains two diffusion models, one per component $z \in \{\boldsymbol{\tau}, \mathbf{r}\}$.

**Forward diffusion (per component).** For each $z \in \{\boldsymbol{\tau}, \mathbf{r}\}$ we use a standard Gaussian forward chain

$$q(z_{1:S} \mid z_0) = \prod_{t=1}^{S} q(z_t \mid z_{t-1}), \qquad q(z_t \mid z_{t-1}) = \mathcal{N}(\sqrt{\alpha_t}\, z_{t-1},\, (1 - \alpha_t)\, \mathbf{I}), \tag{20}$$

with $\bar{\alpha}_t = \prod_{s=1}^{t} \alpha_s$ and

$$z_t = \sqrt{\bar{\alpha}_t}\, z_0 + \sqrt{1 - \bar{\alpha}_t}\, \boldsymbol{\epsilon}, \qquad \boldsymbol{\epsilon} \sim \mathcal{N}(\mathbf{0}, \mathbf{I}),$$

where $\mathbf{I}$ denotes the identity of appropriate dimension (i.e., matching the vectorized $z_t$).

**Reverse model (per component).** The reverse (generative) model is

$$p_\theta(z_{0:S}) \;=\; p(z_S) \prod_{t=1}^{S} p_\theta(z_{t-1} \mid z_t), \qquad p(z_S) \;=\; \mathcal{N}(\mathbf{0}, \mathbf{I}), \tag{21}$$

with $p_\theta(z_{t-1} \mid z_t)$ parameterized via a noise-prediction network $\boldsymbol{\epsilon}_\theta(z_t, t, \text{cond})$. The usual DDPM objective reduces to

$$\mathcal{L}_z^{\text{simple}}(\theta) \;=\; \mathbb{E}_{z_0, t, \boldsymbol{\epsilon}}\Big[w_t \left\| \boldsymbol{\epsilon} - \boldsymbol{\epsilon}_\theta(z_t, t, \text{cond}) \right\|_2^2\Big] \quad \text{(up to $t$-dependent constants).} \tag{22}$$

In DiffPM, "cond" may include positional metadata (e.g., window start and length); we leave it implicit here to focus on likelihood.

### B.2   Joint generative density and variational family

The *observable* is always recovered at $t = 0$ by $\mathbf{x}_0 = \boldsymbol{\tau}_0 + \mathbf{r}_0$. We use $c$ to denote window context/metadata: start index $s$, total series length $T$, and any positional metadata consumed by the denoiser. All likelihoods/ELBOs in this appendix are conditional on $c$ unless stated otherwise. DiffPM uses *context-conditional independence* of trend and residual given $c$:

1. **Independent components (used in DiffPM):** $p_\theta(\boldsymbol{\tau}_0, \mathbf{r}_0 \mid c) \;=\; p_{\theta_1}(\boldsymbol{\tau}_0 \mid c)\, p_{\theta_2}(\mathbf{r}_0 \mid c)$.
2. **Hierarchical :** $p_\theta(\boldsymbol{\tau}_0, \mathbf{r}_0 \mid c) \;=\; p_{\theta_1}(\boldsymbol{\tau}_0 \mid c)\, p_{\theta_2}(\mathbf{r}_0 \mid \boldsymbol{\tau}_0, c)$.

Let $\delta(\cdot)$ denote a Dirac delta. The joint *generative* density over reverse paths and the observation is

$$p_\theta(\boldsymbol{\tau}_{0:S}, \mathbf{r}_{0:S}, \mathbf{x}_0 \mid c) \;=\; p_{\theta_1}(\boldsymbol{\tau}_{0:S} \mid c)\, p_{\theta_2}(\mathbf{r}_{0:S} \mid \boldsymbol{\tau}_0, c)\, \delta(\mathbf{x}_0 - \boldsymbol{\tau}_0 - \mathbf{r}_0), \tag{23}$$

where in the independent case $p_{\theta_2}(\mathbf{r}_{0:S} \mid \boldsymbol{\tau}_0, c)$ reduces to $p_{\theta_2}(\mathbf{r}_{0:S} \mid c)$.

For training, we use the *forward* variational family that fixes the decomposition at $t = 0$ and then applies forward diffusion:

$$q(\boldsymbol{\tau}_{0:S}, \mathbf{r}_{0:S} \mid \mathbf{x}_0, c) \;=\; \underbrace{\delta\big(\boldsymbol{\tau}_0 - \mathcal{T}(\mathbf{x}_0)\big)\, \delta\big(\mathbf{r}_0 - (\mathbf{x}_0 - \mathcal{T}(\mathbf{x}_0))\big)}_{\text{deterministic decomposition at } t=0} \left[\prod_{t=1}^{S} q(\boldsymbol{\tau}_t \mid \boldsymbol{\tau}_{t-1})\right]\left[\prod_{t=1}^{S} q(\mathbf{r}_t \mid \mathbf{r}_{t-1}, \boldsymbol{\tau}_0)\right], \tag{24}$$

with $q(\mathbf{r}_t \mid \mathbf{r}_{t-1}, \boldsymbol{\tau}_0) = q(\mathbf{r}_t \mid \mathbf{r}_{t-1})$ in the independent case.

The ELBO on $\ln p_\theta(\mathbf{x}_0 \mid c)$ is

$$\mathcal{L}(\mathbf{x}_0 \mid c) \;=\; \mathbb{E}_q\left[\ln \frac{p_\theta(\boldsymbol{\tau}_{0:S}, \mathbf{r}_{0:S}, \mathbf{x}_0 \mid c)}{q(\boldsymbol{\tau}_{0:S}, \mathbf{r}_{0:S} \mid \mathbf{x}_0, c)}\right] \;\le\; \ln p_\theta(\mathbf{x}_0 \mid c). \tag{25}$$

Because $q$ puts $(\boldsymbol{\tau}_0, \mathbf{r}_0)$ exactly on the support of $\delta(\mathbf{x}_0 - \boldsymbol{\tau}_0 - \mathbf{r}_0)$, the delta cancels in the ratio and, under the independent model, the expectation factorizes into a sum of the two component-wise diffusion ELBOs (each conditioned on $c$).

### B.3   Independent components: ELBO splits into two diffusion ELBOs

Under the context-conditional independence used in DiffPM, $p_{\theta_2}(\mathbf{r}_{0:S} \mid \boldsymbol{\tau}_0, c) \equiv p_{\theta_2}(\mathbf{r}_{0:S} \mid c)$, the master ELBO equation 25 reduces to a sum of two standard diffusion ELBOs, one per branch:

$$\mathcal{L}_{\text{indep}}(\mathbf{x}_0 \mid c) = \mathbb{E}_{q(\boldsymbol{\tau}_{1:S} \mid \boldsymbol{\tau}_0)}\left[\ln \frac{p(\boldsymbol{\tau}_S)}{q(\boldsymbol{\tau}_S \mid \boldsymbol{\tau}_0)} + \sum_{t=1}^{S} \ln \frac{p_{\theta_1}(\boldsymbol{\tau}_{t-1} \mid \boldsymbol{\tau}_t, c)}{q(\boldsymbol{\tau}_t \mid \boldsymbol{\tau}_{t-1})}\right] \tag{26}$$

$$+ \mathbb{E}_{q(\mathbf{r}_{1:S} \mid \mathbf{r}_0)}\left[\ln \frac{p(\mathbf{r}_S)}{q(\mathbf{r}_S \mid \mathbf{r}_0)} + \sum_{t=1}^{S} \ln \frac{p_{\theta_2}(\mathbf{r}_{t-1} \mid \mathbf{r}_t, c)}{q(\mathbf{r}_t \mid \mathbf{r}_{t-1})}\right]. \tag{27}$$

Each bracket in equation 26–equation 27 is the usual diffusion ELBO (conditioned on $c$) for its component. Denoting their negative ELBOs by $L_\tau(\boldsymbol{\tau}_0 \mid c)$ and $L_r(\mathbf{r}_0 \mid c)$, we obtain the conditional bound

$$-\ln p_\theta(\mathbf{x}_0 \mid c) \;\le\; L_\tau(\boldsymbol{\tau}_0 \mid c) \;+\; L_r(\mathbf{r}_0 \mid c). \tag{28}$$

**Simple $\epsilon$-prediction form.** Using the standard $\epsilon$-parameterization of the reverse kernels, each branch admits the DDPM "simple" loss (up to $t$-dependent constants):

$$L_\tau(\boldsymbol{\tau}_0 \mid c) \;=\; \mathbb{E}_{\boldsymbol{\tau}_0,\, t,\, \boldsymbol{\epsilon}}\Big[w_t \left\|\boldsymbol{\epsilon} - \boldsymbol{\epsilon}_{\theta_1}(\boldsymbol{\tau}_t, t, c)\right\|_2^2\Big], \qquad L_r(\mathbf{r}_0 \mid c) \;=\; \mathbb{E}_{\mathbf{r}_0,\, t,\, \boldsymbol{\epsilon}}\Big[w_t \left\|\boldsymbol{\epsilon} - \boldsymbol{\epsilon}_{\theta_2}(\mathbf{r}_t, t, c)\right\|_2^2\Big], \tag{29}$$

where $\boldsymbol{\tau}_t = \sqrt{\bar{\alpha}_t}\,\boldsymbol{\tau}_0 + \sqrt{1 - \bar{\alpha}_t}\,\boldsymbol{\epsilon}$ and $\mathbf{r}_t = \sqrt{\bar{\alpha}_t}\,\mathbf{r}_0 + \sqrt{1 - \bar{\alpha}_t}\,\boldsymbol{\epsilon}$, with $\boldsymbol{\epsilon} \sim \mathcal{N}(\mathbf{0}, \mathbf{I})$ and $w_t$ any chosen weighting (e.g., uniform or SNR-based). *Unless otherwise stated, noise draws are i.i.d. across time, channels, windows, and branches; we reuse the symbol $\boldsymbol{\epsilon}$ for notational brevity.*

**Training objective.** Summing the two component losses yields the DiffPM training criterion

$$\mathcal{J}_{\text{indep}}(\theta_1, \theta_2) \;=\; \mathbb{E}_{c \sim \hat{p}(C)}\Big[\mathbb{E}\big[w_t \|\boldsymbol{\epsilon} - \boldsymbol{\epsilon}_{\theta_1}(\boldsymbol{\tau}_t, t, c)\|_2^2\big] + \mathbb{E}\big[w_t \|\boldsymbol{\epsilon} - \boldsymbol{\epsilon}_{\theta_2}(\mathbf{r}_t, t, c)\|_2^2\big]\Big] \quad \text{(constants omitted)}, \tag{30}$$

i.e., the expectation over window contexts $c$ (under the empirical distribution of contexts in the dataset) of the sum of per-branch $\epsilon$-MSE objectives. Equation equation 28 then guarantees that minimizing equation 30 minimizes an upper bound on the conditional negative log-likelihood $-\ln p_\theta(\mathbf{x}_0 \mid c)$.

## B.4 HIERARCHICAL (CONDITIONAL) COMPONENTS: ELBO SPLITS INTO TREND + CONDITIONAL RESIDUAL

The hierarchical variant assumes the residual distribution depends on the realized trend (in addition to window context $c$):

$$p_\theta(\boldsymbol{\tau}_0, \mathbf{r}_0 \mid c) \;=\; p_{\theta_1}(\boldsymbol{\tau}_0 \mid c)\, p_{\theta_2}(\mathbf{r}_0 \mid \boldsymbol{\tau}_0, c).$$

(We include this as an *optional extension*; DiffPM's main experiments use the independent model in Section B.3.)

Under this factorization, the master ELBO equation 25 decomposes into a sum of two diffusion ELBOs: one for the marginal trend and one for the *conditional* residual. Writing expectations under the forward processes (with the deterministic decomposition at $t = 0$), we obtain

$$\mathcal{L}_{\text{hier}}(\mathbf{x}_0 \mid c) = \underbrace{\mathbb{E}_{q(\boldsymbol{\tau}_{1:S} \mid \boldsymbol{\tau}_0)}\Big[\ln \frac{p(\boldsymbol{\tau}_S)}{q(\boldsymbol{\tau}_S \mid \boldsymbol{\tau}_0)} + \sum_{t=1}^{S} \ln \frac{p_{\theta_1}(\boldsymbol{\tau}_{t-1} \mid \boldsymbol{\tau}_t, c)}{q(\boldsymbol{\tau}_t \mid \boldsymbol{\tau}_{t-1})}\Big]}_{\text{ELBO for } p_{\theta_1}(\boldsymbol{\tau}_0 \mid c)} \tag{31}$$

$$+ \underbrace{\mathbb{E}_{q(\mathbf{r}_{1:S} \mid \mathbf{r}_0, \boldsymbol{\tau}_0)}\Big[\ln \frac{p(\mathbf{r}_S)}{q(\mathbf{r}_S \mid \mathbf{r}_0, \boldsymbol{\tau}_0)} + \sum_{t=1}^{S} \ln \frac{p_{\theta_2}(\mathbf{r}_{t-1} \mid \mathbf{r}_t, \boldsymbol{\tau}_0, c)}{q(\mathbf{r}_t \mid \mathbf{r}_{t-1}, \boldsymbol{\tau}_0)}\Big]}_{\text{ELBO for } p_{\theta_2}(\mathbf{r}_0 \mid \boldsymbol{\tau}_0, c)}. \tag{32}$$

Denote the corresponding negative ELBOs by $L_\tau(\boldsymbol{\tau}_0 \mid c)$ and $L_{r\mid\tau}(\mathbf{r}_0 \mid \boldsymbol{\tau}_0, c)$. Then the conditional NLL is upper-bounded by their sum:

$$-\ln p_\theta(\mathbf{x}_0 \mid c) \;\leq\; L_\tau(\boldsymbol{\tau}_0 \mid c) \;+\; L_{r\mid\tau}(\mathbf{r}_0 \mid \boldsymbol{\tau}_0, c). \tag{33}$$

**Simple $\epsilon$-prediction form.** With the standard $\epsilon$-parameterization, each term admits the DDPM simple loss (up to $t$-dependent constants):

$$L_\tau(\boldsymbol{\tau}_0 \mid c) = \mathbb{E}_{\boldsymbol{\tau}_0,\, t,\, \boldsymbol{\epsilon}}\Big[w_t \left\|\boldsymbol{\epsilon} - \boldsymbol{\epsilon}_{\theta_1}(\boldsymbol{\tau}_t, t, c)\right\|_2^2\Big], \qquad L_{r\mid\tau}(\mathbf{r}_0 \mid \boldsymbol{\tau}_0, c) = \mathbb{E}_{\mathbf{r}_0,\, t,\, \boldsymbol{\epsilon}}\Big[w_t \left\|\boldsymbol{\epsilon} - \boldsymbol{\epsilon}_{\theta_2}(\mathbf{r}_t, t, \boldsymbol{\tau}_0, c)\right\|_2^2\Big], \tag{34}$$

where $\boldsymbol{\tau}_t = \sqrt{\bar{\alpha}_t}\,\boldsymbol{\tau}_0 + \sqrt{1 - \bar{\alpha}_t}\,\boldsymbol{\epsilon}$ and $\mathbf{r}_t = \sqrt{\bar{\alpha}_t}\,\mathbf{r}_0 + \sqrt{1 - \bar{\alpha}_t}\,\boldsymbol{\epsilon}$, with $\boldsymbol{\epsilon} \sim \mathcal{N}(\mathbf{0}, \mathbf{I})$ and $w_t$ a chosen weighting (e.g., uniform or SNR-based).

**Training objective.** Summing the two branches yields the hierarchical training criterion

$$\mathcal{J}_{\text{hier}}(\theta_1, \theta_2) \;=\; \mathbb{E}_{c \sim \hat{p}(C)}\Big[\mathbb{E}\big[w_t \|\boldsymbol{\epsilon} - \boldsymbol{\epsilon}_{\theta_1}(\boldsymbol{\tau}_t, t, c)\|_2^2\big] + \mathbb{E}\big[w_t \|\boldsymbol{\epsilon} - \boldsymbol{\epsilon}_{\theta_2}(\mathbf{r}_t, t, \boldsymbol{\tau}_0, c)\|_2^2\big]\Big] \quad \text{(constants omitted)}. \tag{35}$$

Equations equation 33–equation 35 show that, even when the residual is explicitly conditioned on the realized trend, the overall objective remains a *sum of diffusion objectives*: one marginal (trend) and one conditional (residual). This preserves the clean additive structure used throughout our windowed ELBO, and reduces to Section B.3 when the conditional dependence is removed.

## B.5 FROM FULL SEQUENCES TO WINDOWS

Let a long sequence be split into $M$ training windows $\{\mathbf{x}_0^{(i)}\}_{i=1}^M$ with deterministic decompositions $\mathbf{x}_0^{(i)} = \boldsymbol{\tau}_0^{(i)} + \mathbf{r}_0^{(i)}$ as in equation 19. Let $c_i$ denote window metadata (start index $s_i$ and total series length $T$) for window $i$.

**Non-overlapping windows.** Suppose the windows form a partition of the sequence (no shared time indices). Under the context-conditional independent model used in DiffPM,

$$p_\theta(\boldsymbol{\tau}_0^{(i)}, \mathbf{r}_0^{(i)} \mid c_i) = p_{\theta_1}(\boldsymbol{\tau}_0^{(i)} \mid c_i)\, p_{\theta_2}(\mathbf{r}_0^{(i)} \mid c_i),$$

the conditional negative log-likelihood (NLL) of the full sequence given the set $C = \{c_i\}_{i=1}^M$ satisfies the additive ELBO bound

$$-\ln p_\theta(\mathbf{x}_{\text{full}} \mid C) \;\le\; \sum_{i=1}^M \Big( L_\tau(\boldsymbol{\tau}_0^{(i)} \mid c_i) + L_r(\mathbf{r}_0^{(i)} \mid c_i) \Big), \qquad \text{(independent model)} \qquad (36)$$

where $L_\tau$ and $L_r$ are the component-wise diffusion NELBOs (each conditioned on $c_i$) from equation 29. In the hierarchical extension of Section B.4, the bound becomes $\sum_i \big( L_\tau(\boldsymbol{\tau}_0^{(i)} \mid c_i) + L_{r|\tau}(\mathbf{r}_0^{(i)} \mid \boldsymbol{\tau}_0^{(i)}, c_i) \big)$. *Equality* holds in equation 36 if the generative model factorizes across windows given $C$, i.e., $p_\theta(\mathbf{x}_{\text{full}} \mid C) = \prod_i p_\theta(\mathbf{x}_0^{(i)} \mid c_i)$; otherwise, the right-hand side is a standard blockwise ELBO surrogate. Equation equation 36 matches the standard blockwise diffusion objective: disjoint blocks contribute additively to the global variational bound.

**Overlapping windows: notation.** Let $\mathcal{T}$ index discrete time points of the full sequence. For window $i$, let $\mathcal{T}_i \subset \mathcal{T}$ denote the set of indices covered by that window, and let $n_t = \sum_{i=1}^M \mathbf{1}\{t \in \mathcal{T}_i\}$ be the number of windows that cover time $t$ (we reuse $t$ for sequence indices in $\mathcal{T}$; context disambiguates diffusion step vs. sequence time). *Hereafter, $t$ inside $q(\cdot)$, $\bar{\alpha}_t$, and $w_t$ denotes the diffusion step, while $t \in \mathcal{T}$ denotes a sequence index.* Define per-window *inclusion weights* $\gamma_t^{(i)} \in [0, 1]$ such that

$$\sum_{i:\, t \in \mathcal{T}_i} \gamma_t^{(i)} \;=\; 1 \qquad \text{for all } t \in \mathcal{T}. \tag{37}$$

Two common choices are: (i) uniform sharing, $\gamma_t^{(i)} = 1/n_t$ for all $i$ with $t \in \mathcal{T}_i$; (ii) tapering (e.g., linear or Hann) within overlaps, still normalized to respect equation 43. We write $\boldsymbol{\Gamma}^{(i)}$ for the diagonal mask with entries $\gamma_t^{(i)}$ restricted to $\mathcal{T}_i$.

A. INCLUSION–EXCLUSION REWEIGHTING (NO DOUBLE COUNTING).

A principled surrogate for the conditional sequence NLL is obtained by *redistributing* overlap mass according to $\gamma_t^{(i)}$. Concretely, define the window-level per-time NELBO contributions $\ell_\tau^{(i)}(t)$ and $\ell_r^{(i)}(t)$ so that $L_\tau(\boldsymbol{\tau}_0^{(i)} \mid c_i) = \sum_{t \in \mathcal{T}_i} \ell_\tau^{(i)}(t)$ and similarly for $L_r$. The overlap-aware objective is

$$\widetilde{\mathcal{L}} \;=\; \sum_{i=1}^M \sum_{t \in \mathcal{T}_i} \gamma_t^{(i)} \Big( \ell_\tau^{(i)}(t) + \ell_r^{(i)}(t) \Big), \tag{38}$$

which can be equivalently written in vector form as $\sum_i \langle \boldsymbol{\Gamma}^{(i)}, \ell_\tau^{(i)} + \ell_r^{(i)} \rangle$ with inner product over the time axis of window $i$. By construction equation 43 implies that each global time index contributes *exactly once* to equation 38, so there is no double counting. Replacing $\ell^{(i)}(t)$ with the per-time DDPM simple losses (up to constants) yields a practical training loss that coincides with the usual per-window objective when windows do not overlap ($\gamma_t^{(i)} \equiv 1$). In our implementation, training uses framewise $\epsilon$-MSE, so the per-time decomposition used here is exact.

B. CONDITIONAL (BLOCKWISE) CHAINING WITH OVERLAP CONSISTENCY.

An alternative view generates windows in a fixed order while enforcing agreement on overlaps. Let $i = 1, \ldots, M$ index the order (e.g., increasing start times). Write $\mathbf{o}^{(i)}$ for the *observed* overlap

values of window $i$ that are already committed by previously generated windows, and let $\mathcal{O}^{(i)} \subset \mathcal{T}_i$ be the corresponding time indices. Define the *conditional* per-window generative factors

$$p_{\theta_1}^{(i)}\Big(\boldsymbol{\tau}_0^{(i)} \mid c_i, \mathbf{o}^{(i)}\Big) \propto \int p_{\theta_1}(\boldsymbol{\tau}_{0:S}^{(i)} \mid c_i) \prod_{t \in \mathcal{O}^{(i)}} \delta\big(\tau_{0,t}^{(i)} - o_t^{(i)}\big) \, d\boldsymbol{\tau}_{1:S}^{(i)}, \tag{39}$$

$$p_{\theta_2}^{(i)}\Big(\mathbf{r}_0^{(i)} \mid c_i, \mathbf{o}^{(i)}\Big) \propto \int p_{\theta_2}(\mathbf{r}_{0:S}^{(i)} \mid c_i) \prod_{t \in \mathcal{O}^{(i)}} \delta\big(r_{0,t}^{(i)} - \tilde{o}_t^{(i)}\big) \, d\mathbf{r}_{1:S}^{(i)}, \tag{40}$$

where $(o_t^{(i)}, \tilde{o}_t^{(i)})$ encode the trend/residual parts of the overlap constraint. Because the decomposition is deterministic, the committed overlap on $\mathbf{x}$ induces unique $(o_t^{(i)}, \tilde{o}_t^{(i)})$ via $(o_t^{(i)}, \tilde{o}_t^{(i)}) = (\mathcal{T}(\mathbf{x})_t, \ \mathbf{x}_t - \mathcal{T}(\mathbf{x})_t)$ on $\mathcal{O}^{(i)}$. The conditional ELBO for window $i$ then reads

$$\mathcal{L}_{\mathrm{cond}}^{(i)}(\mathbf{x}_0^{(i)} \mid c_i, \mathbf{o}^{(i)}) \ \le \ \ln p_\theta\Big(\mathbf{x}_0^{(i)} \mid c_i, \mathbf{o}^{(i)}\Big), \tag{41}$$

and the global conditional NLL obeys the additive bound

$$-\ln p_\theta(\mathbf{x}_{\mathrm{full}} \mid C) \ \le \ \sum_{i=1}^{M} \Big(L_\tau^{(i)}(\boldsymbol{\tau}_0^{(i)} \mid c_i, \mathbf{o}^{(i)}) + L_r^{(i)}(\mathbf{r}_0^{(i)} \mid c_i, \mathbf{o}^{(i)})\Big), \tag{42}$$

with $L^{(i)}$ the conditional NELBOs corresponding to equation 41. In practice, imposing $\mathbf{o}^{(i)}$ is implemented by *clamping* the overlap entries to their committed values during the forward/backward passes of the DDPM objective for window $i$ (equivalently, by zeroing the loss on clamped coordinates; we supply both the binary mask $\mathbf{M}_i$ and the clamped overlap values $z_0^{(i)} \upharpoonright \mathcal{O}_{i-1,i}$ as conditioning features). This "blockwise chaining" recovers the non-overlap result when $\mathcal{O}^{(i)} = \varnothing$ and preserves coherence across window boundaries when overlaps are present.

**Remarks.** (i) Objectives equation 38 and equation 42 are complementary. The former is *embarrassingly parallel* (no dependency across windows) and avoids double counting via normalized weights; the latter is *sequential* but enforces hard consistency on overlaps. (ii) Both reduce to equation 36 when windows are disjoint. (iii) Either construction is compatible with the independent model (used in DiffPM) or the hierarchical extension by replacing $L_r$ with $L_{r|\tau}$ as in Section B.4.

### B.6 Overlapping windows I: inclusion–exclusion (reweighting)

Let the full sequence be indexed by a discrete set $\mathcal{T}$. Window $i \in \{1, \dots, M\}$ covers a subset $\mathcal{T}_i \subset \mathcal{T}$ (its *support*). For two consecutive windows $i$ and $i+1$, the overlap index set is $\mathcal{O}_{i,i+1} = \mathcal{T}_i \cap \mathcal{T}_{i+1}$ (of size $O$). For general sliding windows, define the *coverage multiplicity* of time index $t$ as

$$n_t \ = \ \big|\{\, i : t \in \mathcal{T}_i \,\}\big| \ \in \ \{1, 2, \dots\}.$$

To avoid double counting when summing window losses, assign *per-index weights* $\gamma_t^{(i)} \in [0, 1]$ that satisfy the local normalization

$$\sum_{i \,:\, t \in \mathcal{T}_i} \gamma_t^{(i)} \ = \ 1, \qquad \forall\, t \in \mathcal{T}. \tag{43}$$

A convenient choice is uniform sharing, $\gamma_t^{(i)} = 1/n_t$ for all $i$ with $t \in \mathcal{T}_i$. Let $c_i$ denote the metadata (start $s_i$, series length $T$) for window $i$ and let the (per-window) diffusion NELBO decompose over time indices,[2]

$$L^{(i)}\big(\mathbf{x}_0^{(i)} \mid c_i\big) \ \approx \ \sum_{t \in \mathcal{T}_i} \Big(\ell_\tau^{(i)}(t \mid c_i) \ + \ \ell_r^{(i)}(t \mid c_i)\Big). \tag{44}$$

Then the *reweighted* global objective

$$\widetilde{\mathcal{L}} \ = \ \sum_{i=1}^{M} \sum_{t \in \mathcal{T}_i} \gamma_t^{(i)} \Big(\ell_\tau^{(i)}(t \mid c_i) + \ell_r^{(i)}(t \mid c_i)\Big) \tag{45}$$

---

[2]This decomposition is exact for objectives that sum over time (e.g., framewise $\epsilon$-MSE); it is a standard approximation otherwise and works well in practice.

counts each time index *exactly once* by equation 43. For the common case of fixed window length $L$, fixed overlap size $O$, and stride $S = L - O$, we have $n_t = 2$ inside overlaps and $n_t = 1$ elsewhere, so $\gamma_t^{(i)} = \frac{1}{2}$ on overlaps and 1 elsewhere—equivalent to subtracting one copy of the overlap from the naïve sum (an inclusion–exclusion correction). Formally, for two windows $W_1, W_2$,

$$\ln p_\theta(W_1 \cup W_2 \mid c_1, c_2) \;=\; \ln p_\theta(W_1 \mid c_1) \;+\; \ln p_\theta(W_2 \mid c_2) \;-\; \ln p_\theta(W_1 \cap W_2 \mid c_1, c_2), \quad (46)$$

and the reweighted objective mirrors this identity at the ELBO level by distributing the overlap mass.

**Component-wise view.** The same weights apply to the trend and residual branches individually:

$$\widetilde{\mathcal{L}} \;=\; \sum_{i=1}^{M} \sum_{t \in \mathcal{T}_i} \gamma_t^{(i)} \, \ell_\tau^{(i)}(t \mid c_i) \;+\; \sum_{i=1}^{M} \sum_{t \in \mathcal{T}_i} \gamma_t^{(i)} \, \ell_r^{(i)}(t \mid c_i), \quad (47)$$

so the two-branch structure is preserved under reweighting. In the hierarchical extension (Section B.4), replace $\ell_r^{(i)}$ by its conditional counterpart $\ell_{r|\tau}^{(i)}$.

### B.7 OVERLAPPING WINDOWS II: CONDITIONAL (BLOCKWISE) CHAINING

An alternative is to make each window *conditional* on the overlap with already-committed windows, so no double counting arises in the first place. Let $\mathbf{W}_i$ be the $i$-th window in a fixed order (e.g., increasing start indices). Write $\mathcal{O}_{i-1,i} = \mathcal{T}_{i-1} \cap \mathcal{T}_i$ and decompose $\mathbf{W}_i = (\mathbf{W}_i^{\mathrm{ov}}, \mathbf{W}_i^{\mathrm{new}})$ where $\mathbf{W}_i^{\mathrm{ov}}$ is known from $\mathbf{W}_{i-1}$ on $\mathcal{O}_{i-1,i}$. Conditioning on the set of window contexts $C = \{c_i\}_{i=1}^{M}$, the sequence likelihood factorizes as

$$p_\theta(\mathbf{x}_{\mathrm{full}} \mid C) \;=\; p_\theta(\mathbf{W}_1 \mid c_1) \prod_{i=2}^{M} p_\theta\big(\mathbf{W}_i^{\mathrm{new}} \,\big|\, \mathbf{W}_{i-1} \!\upharpoonright \mathcal{O}_{i-1,i}, \, c_i\big). \quad (48)$$

Taking $-\ln$ and applying ELBOs termwise yields a *sum of conditional ELBOs*:

$$-\ln p_\theta(\mathbf{x}_{\mathrm{full}} \mid C) \;\le\; L(\mathbf{W}_1 \mid c_1) \;+\; \sum_{i=2}^{M} L\Big(\mathbf{W}_i^{\mathrm{new}} \,\Big|\, \mathbf{W}_{i-1} \!\upharpoonright \mathcal{O}_{i-1,i}, \, c_i\Big), \quad (49)$$

where each $L(\cdot)$ is the (conditional) diffusion NELBO for the corresponding factor.

**Diffusion implementation via clamping (mask/inpainting).** For branch $z \in \{\boldsymbol{\tau}, \mathbf{r}\}$, construct a binary mask $\mathbf{M}_i \in \{0, 1\}^{L \times D}$ that is 1 on the overlap indices $\mathcal{O}_{i-1,i}$ of window $i$ and 0 elsewhere. Given clean $z_0^{(i)}$ and its noised version $z_t^{(i)} = \sqrt{\bar{\alpha}_t} \, z_0^{(i)} + \sqrt{1 - \bar{\alpha}_t} \, \boldsymbol{\epsilon}$, form the *clamped* input

$$\tilde{z}_t^{(i)} \;=\; \mathbf{M}_i \odot z_0^{(i)} \;+\; (1 - \mathbf{M}_i) \odot z_t^{(i)}, \quad (50)$$

where $\odot$ denotes the Hadamard (element-wise) product, so that overlap entries remain fixed to the committed values, while the remaining entries are noised as usual. Train with the masked $\epsilon$-loss

$$\mathcal{L}_{z,i}^{\mathrm{cond}} \;=\; \mathbb{E}\Big[w_t \big\|(1 - \mathbf{M}_i) \odot \big(\boldsymbol{\epsilon} - \boldsymbol{\epsilon}_\theta(\tilde{z}_t^{(i)}, \, t, \, c_i, \, \mathbf{M}_i, \, z_0^{(i)} \!\upharpoonright \mathcal{O}_{i-1,i})\big)\big\|_2^2\Big], \quad (51)$$

which enforces that the model *cannot* change overlap entries and must denoise only the new region. Summing the two branches gives the total conditional objective

$$\mathcal{J}_{\mathrm{cond}} \;=\; \sum_{i=1}^{M} \Big(\mathcal{L}_{\boldsymbol{\tau},i}^{\mathrm{cond}} + \mathcal{L}_{\mathbf{r},i}^{\mathrm{cond}}\Big) \quad \text{(constants omitted)}. \quad (52)$$

**Remarks.** (i) Compared to inclusion–exclusion equation 38, the conditional chaining bound equation 49 enforces boundary coherence by construction (no cross-fading needed) and aligns training with sequential generation. (ii) When overlaps are empty ($\mathcal{O}_{i-1,i} = \varnothing$), the mask is $\mathbf{M}_i \equiv 0$ and equation 51 reduces to the usual per-window DDPM loss. (iii) The construction is compatible with the independent model (used in DiffPM) and with the hierarchical extension by augmenting the residual denoiser input with $\boldsymbol{\tau}_0$ when desired.

## B.8 Assumptions, edge cases, and practical guidance

**Deterministic decomposition.** Equation equation 19 assumes a fixed analysis operator $\mathcal{T}$ (no learned posterior). Hence $q(\boldsymbol{\tau}_0, \mathbf{r}_0 \mid \mathbf{x}_0, c)$ is a product of Diracs that exactly satisfy $\mathbf{x}_0 = \boldsymbol{\tau}_0 + \mathbf{r}_0$, and the Dirac constraint cancels from the ELBO ratio in equation 25. This keeps the variational family simple and makes the ELBO algebra in Sections B.3–B.4 exact up to the usual DDPM constants.

**Independent vs. hierarchical.** If residual statistics depend on the realized trend (e.g., level-dependent variance or trend-modulated periodicity), the hierarchical objective equation 33 is strictly more expressive and can capture such couplings. If not, the context-conditional independent objective equation 28 is simpler, decouples training across branches, and is what DiffPM uses in the main experiments.

**Context and conditioning.** All sequence/window likelihoods and ELBOs are understood *conditional on context $c$*: window start $s$, total series length $T$, and any positional metadata (cf. equation 22 and equation 29). In the hierarchical variant, the residual branch may additionally consume the *realized* clean trend $\boldsymbol{\tau}_0$ as input to the denoiser (see equation 34).

**Per-time decomposition.** The inclusion–exclusion objective equation 38 uses a per-time decomposition equation 44. This is *exact* for framewise DDPM losses (e.g., $\epsilon$-MSE summed over time and channels) and is a standard, accurate approximation otherwise. With fixed window length $L$ and stride $S$, set $\gamma_t^{(i)} = 1$ outside overlaps and $\gamma_t^{(i)} = \frac{1}{2}$ on overlaps. For nonuniform coverages, use the normalized rule $\gamma_t^{(i)} = 1/n_t$, where $n_t$ is the coverage multiplicity (Section B.6).

**Overlap choices and masks.** In the conditional chaining scheme (Section B.7), a single binary mask $\mathbf{M}_i$ is applied to both branches, clamping the overlap entries to already-committed values during training via equation 50–equation 51. The conditioning features can be chosen minimally (raw overlap values), augmented (e.g., learned embeddings of the overlap region), and always include the positional metadata $c_i$.

**Variable lengths and boundaries.** If windows near the sequence boundaries are shorter than $L$ (e.g., due to causal padding or dataset truncation), one can either (i) pad them and keep equation 38 with the same $\gamma_t^{(i)}$, or (ii) shrink $\mathcal{T}_i$ and renormalize weights to satisfy equation 43. Both preserve the additive ELBO structure and avoid double counting.

**Coverage multiplicities $> 2$.** When more than two windows overlap at a time index (e.g., small strides), the inclusion–exclusion normalization equation 43 still applies with $\gamma_t^{(i)} = 1/n_t$, and equation 38 continues to count each index exactly once. Conditional chaining also generalizes by conditioning each new window on *all* already-committed overlaps in $\mathcal{T}_i$.

**Missing values in training windows.** If a training window contains missing entries, either (i) exclude those coordinates from the per-time loss by multiplying equation 29 with a data-mask (equivalently, set $\gamma_t^{(i)} = 0$ at missing indices in equation 38), or (ii) treat them as known inpainting masks within the conditional-chaining view (using $\mathbf{M}_i$ to clamp observed entries and ignore missing ones). Both choices are compatible with the ELBO derivations.

**Training objective summary.** For a dataset of (possibly overlapping) windows with contexts $\{c_i\}_{i=1}^M$, the DiffPM objective is a *sum of branchwise diffusion ELBOs* across windows:

$$\min_{\theta_1, \theta_2} \sum_{i=1}^M \left\{ \mathbb{E}\left[ w_t \left\| \boldsymbol{\epsilon} - \boldsymbol{\epsilon}_{\theta_1}(\boldsymbol{\tau}_t^{(i)}, t, c_i) \right\|_2^2 \right] + \mathbb{E}\left[ w_t \left\| \boldsymbol{\epsilon} - \boldsymbol{\epsilon}_{\theta_2}(\mathbf{r}_t^{(i)}, t, c_i) \right\|_2^2 \right] \right\} \quad \text{(constants omitted)},$$

$$(53)$$

with either

(a) *inclusion–exclusion reweighting*: multiply the per-time losses inside the expectations in equation 53 by the index weights $\gamma_t^{(i)}$ from equation 43, yielding the global reweighted objective equation 38; or

(b) *conditional chaining*: replace the inputs by the clamped $\tilde{z}_t^{(i)}$ from equation 50 and restrict the error to $(1 - \mathbf{M}_i)$ as in equation 51, summing the conditional losses across windows as in equation 52.

Both strategies realize a coherent, additive ELBO over windows and branches, differing only in whether overlap consistency is achieved statistically (reweighting) or enforced as a hard constraint (clamping).

**Generation-time stitching (context only).** Under inclusion–exclusion training, overlapping windows can be merged by overlap-add or linear cross-fades (with weights consistent with equation 43); under conditional chaining, overlaps are fixed by design, so windows join seamlessly without averaging. (Empirical sampling details are discussed in the main text.)

## C  RECONSTRUCTION FIDELITY GUARANTEES FOR OVERLAP–ADD STITCHING

This appendix provides a detailed theoretical analysis of the overlap–add (OLA) stitching mechanism used in DiffPM, specifically when employing constant overlap–add (COLA) windows. We begin by establishing a rigorous mathematical setup, which serves as a foundation for proving perfect reconstruction (PR) guarantees under ideal, noise-free conditions. Subsequently, we derive both worst-case and stochastic error bounds to characterize performance in the presence of local prediction errors. The analysis is extended to assess the robustness of the method to non-ideal window sums, to handle multivariate signals, and to identify optimal weighting schemes under heteroscedastic local errors. It is important to note that all results presented apply independently to both the trend and residual branches of DiffPM; by linearity, the final output inherits the guarantees established for each branch.

### C.1  NOTATION AND SETUP

Let $x[n] \in \mathbb{R}$ represent a discrete-time univariate signal of length $N$. We fix a window length $L \in \mathbb{N}$ and a hop size $R \in \{1, \ldots, L\}$. The analysis centers on a real-valued, non-negative window function $w[n]$ that is supported on the interval $n \in \{0, \ldots, L-1\}$:

$$w[n] \geq 0, \qquad w[n] = 0 \text{ for } n \notin \{0, \ldots, L-1\}.$$

The signal is segmented into a set of frames indexed by $\mathcal{M} = \{0, 1, \ldots, M-1\}$. The $m$-th frame corresponds to the signal indices

$$\mathcal{I}_m = \{mR, mR+1, \ldots, mR+L-1\}.$$

For analytical simplicity, we assume the signal length is $N = (M-1)R + L$, which ensures that the final frame aligns perfectly with the end of the signal, obviating the need for padding.[3]

The window function $w$ is assumed to satisfy the normalized **constant overlap–add (COLA) property** for the given hop size $R$. This property dictates that the sum of all shifted versions of the window is unity at every point in time:

$$\sum_{m \in \mathbb{Z}} w[n - mR] = 1 \qquad \text{for all } n \in \mathbb{Z}. \tag{54}$$

An equivalent formulation involves defining a *coverage function* $S[n]$ over the signal domain:

$$S[n] := \sum_{m \in \mathcal{M}} w[n - mR]. \tag{55}$$

In this context, the COLA condition is met if $S[n] \equiv 1$ for all $n \in \{0, \ldots, N-1\}$.[4]

---

[3] All results can be readily extended to cases involving truncated or padded boundaries, as briefly discussed in Sec. C.9.

[4] Classical window choices satisfying this property include rectangular windows with $R = L$ (no overlap), and Hann or triangular windows with $R = L/2$ (50% overlap), provided they are appropriately normalized. Our analysis does not presume a specific window shape, only that it satisfies Eq. equation 54.

For each frame $m$, a local synthesis model generates a prediction, denoted $y_m[k]$ for $k \in \{0, \ldots, L-1\}$, which aims to approximate the true signal segment $x[mR + k]$. The final reconstructed signal, $\tilde{x}[n]$, is synthesized via **window-weighted overlap–add**:

$$\tilde{x}[n] = \sum_{m=0}^{M-1} w[n - mR]\, y_m[n - mR], \qquad n = 0, \ldots, N - 1. \tag{56}$$

Here, it is understood that $w[j] = 0$ if $j \notin \{0, \ldots, L - 1\}$. In regions where frames overlap, the COLA property ensures that the weights $\{w[n - mR]\}_m$ form a convex partition (i.e., they are non-negative and sum to 1), meaning that Eq. equation 56 effectively computes a weighted average of the overlapping local predictions.

In the DiffPM architecture, this OLA procedure is applied independently to both the trend and residual branches, producing reconstructions $\tilde{x}^{(\mathrm{T})}$ and $\tilde{x}^{(\mathrm{R})}$. The final output is their sum:

$$\tilde{x}_{\text{total}}[n] = \tilde{x}^{(\mathrm{T})}[n] + \tilde{x}^{(\mathrm{R})}[n]. \tag{57}$$

The analysis that follows is presented for a generic single-branch reconstruction $\tilde{x}$. Due to the linearity of the process, the derived properties apply equally to each branch and, by extension, to their sum.

## C.2 PERFECT RECONSTRUCTION (PR)

A fundamental property of the OLA scheme is that it is lossless if the local predictions are perfectly accurate and the COLA condition holds.

**Theorem C.1** (Perfect Reconstruction under COLA). *Assume the COLA property equation 54 is satisfied and that the per-frame predictions are noise-free, i.e.,*

$$y_m[k] = x[mR + k] \qquad \text{for all } m \in \mathcal{M}, \ k \in \{0, \ldots, L - 1\}. \tag{58}$$

*Then the stitched reconstruction is pointwise identical to the ground truth signal:*

$$\tilde{x}[n] = x[n] \qquad \text{for all } n = 0, \ldots, N - 1.$$

*Proof.* Substituting the noise-free condition equation 58 into the OLA formula equation 56, we observe that for any time index $n$, if a term $w[n - mR]$ is non-zero, it implies $n - mR \in \{0, \ldots, L - 1\}$. Let $k = n - mR$. Then $y_m[n - mR] = y_m[k] = x[mR + k] = x[n]$. This allows us to rewrite the OLA sum as:

$$\tilde{x}[n] = \sum_m w[n - mR]\, x[n] = x[n] \left( \sum_m w[n - mR] \right).$$

By the COLA property equation 54, the term in parentheses is equal to 1, which yields $\tilde{x}[n] = x[n]$. $\qquad \square$

This result has two important implications. First, PR holds for both the trend and residual branches independently. If each branch is reconstructed without error, their sum will perfectly recover the original signal, since the initial decomposition was additive. Second, the proof highlights that PR in the noise-free case is guaranteed by any set of weights that form a partition of unity; the specific window shape is not critical for this property.

## C.3 DETERMINISTIC ERROR ANALYSIS: WORST–CASE BOUNDS

We now analyze the behavior of the reconstruction error when local predictions are imperfect. Let us define the local, per-frame error as

$$e_m[k] := y_m[k] - x[mR + k], \qquad k = 0, \ldots, L - 1. \tag{59}$$

The global reconstruction error, $\tilde{x}[n] - x[n]$, can be expressed in terms of these local errors. By subtracting $x[n]$ from Eq. equation 56 and using the identity $x[n] = \sum_m w[n - mR]\, x[n]$, we find:

$$\tilde{x}[n] - x[n] = \sum_{m=0}^{M-1} w[n - mR]\, e_m[n - mR]. \tag{60}$$

This key equation reveals that the total error at any point $n$ is a convex combination of the local errors from all contributing frames. This structure leads directly to a strong worst-case guarantee.

**Proposition C.2** (Supremum-Norm Guarantee). *If the magnitude of the local error is uniformly bounded by a constant $\varepsilon$, i.e., $|e_m[k]| \leq \varepsilon$ for all $m$ and $k$, then the global reconstruction error is also bounded by $\varepsilon$: $|\tilde{x}[n] - x[n]| \leq \varepsilon$ for all $n$.*

*Proof.* Applying the triangle inequality to Eq. equation 60 yields:

$$|\tilde{x}[n] - x[n]| \ \leq \ \sum_m w[n - mR] \, |e_m[n - mR]| \ \leq \ \sum_m w[n - mR] \, \varepsilon.$$

Since the weights are non-negative and sum to 1 by the COLA property equation 54, the expression simplifies to:

$$|\tilde{x}[n] - x[n]| \ \leq \ \varepsilon \sum_m w[n - mR] \ = \ \varepsilon.$$

$\square$

This proposition establishes that OLA with a COLA window does not amplify the worst-case error; the stitched error's maximum deviation is no greater than the maximum local deviation. Furthermore, in practical scenarios where local errors $e_m[n - mR]$ might have varying signs across overlapping frames, the weighted averaging in Eq. equation 60 can lead to error cancellation, resulting in a global error that is often much smaller than the worst-case bound $\varepsilon$.

C.4 STOCHASTIC ERROR ANALYSIS: UNBIASEDNESS AND VARIANCE

To gain deeper insight, we adopt a stochastic perspective. Assume the local errors are random variables with zero mean and finite variance. For a given absolute time index $n$, let the statistical properties of the error from frame $m$ be:

$$\mathbb{E}\big[e_m[n - mR]\big] = 0, \qquad \mathrm{Var}\big(e_m[n - mR]\big) = \sigma_m^2[n].$$

We also allow for non-zero covariance between errors from different frames, $m \neq m'$, that overlap at time $n$: $\mathrm{Cov}\big(e_m[n - mR], e_{m'}[n - m'R]\big) = \gamma_{m,m'}[n]$.

**Proposition C.3** (Unbiasedness and Variance with General Covariance). *The OLA reconstruction is unbiased, and its variance at each time step $n$ is given by:*

$$\mathbb{E}[\tilde{x}[n] - x[n]] \ = \ 0, \tag{61}$$

$$\mathrm{Var}(\tilde{x}[n] - x[n]) \ = \ \sum_m w[n - mR]^2 \, \sigma_m^2[n] \ + \ 2 \sum_{m<m'} w[n - mR] \, w[n - m'R] \, \gamma_{m,m'}[n]. \tag{62}$$

*Proof.* Unbiasedness follows directly from the linearity of expectation applied to Eq. equation 60: $\mathbb{E}[\tilde{x} - x] = \sum_m w[n - mR] \, \mathbb{E}[e_m[n - mR]] = 0$. The variance formula is obtained by applying the standard formula for the variance of a weighted sum of correlated random variables to Eq. equation 60. $\square$

A particularly important special case arises when local errors are **independent and homoscedastic**. If the errors are uncorrelated across frames at time $n$ ($\gamma_{m,m'}[n] = 0$ for $m \neq m'$) and share a common variance $\sigma^2$, Eq. equation 62 simplifies considerably:

$$\mathrm{Var}(\tilde{x}[n] - x[n]) \ = \ \sigma^2 \sum_m w[n - mR]^2. \tag{63}$$

This result demonstrates a key benefit of overlap-add: variance reduction. Since $\sum_m w[n - mR] = 1$ and $w[n] \geq 0$, the sum of squared weights $\sum_m w[n - mR]^2$ is always less than or equal to 1. For instance, if $K$ frames overlap at time $n$ with roughly equal weights $w \approx 1/K$, then the variance is reduced by a factor of $\sum_m w^2 \approx K(1/K)^2 = 1/K$. A 50% overlap with symmetric weights $(1/2, 1/2)$ would halve the variance.

Conversely, Eq. equation 62 quantifies the impact of **correlated errors**. Positive correlations ($\gamma_{m,m'} > 0$) diminish the variance reduction, as errors tend to reinforce each other. Negative correlations, on the other hand, enhance it through destructive interference. The parallel generation of frames in DiffPM helps ensure that cross-frame error correlations are typically small, allowing the benefits of variance reduction to be realized.

### C.5    ROBUSTNESS TO NON-IDEAL WINDOW SUMS

In practice, the COLA property may hold only approximately, for instance due to boundary effects or floating-point inaccuracies. Let us analyze the impact of such deviations. We use the coverage function $S[n] = \sum_m w[n - mR]$ defined in Eq. equation 55 and assume it is close to unity, bounded as

$$1 - \delta \ \leq \ S[n] \ \leq \ 1 + \delta \qquad \text{for all } n, \tag{64}$$

for some small $\delta \in [0, 1)$. The reconstruction in this case becomes:

$$\tilde{x}[n] \ = \ \sum_m w[n - mR]\,(x[n] + e_m[n - mR]) \ = \ S[n]\,x[n] \ + \ \underbrace{\sum_m w[n - mR]\,e_m[n - mR]}_{\text{error term}}.$$

The total deviation from the true signal is therefore:

$$\tilde{x}[n] - x[n] \ = \ (S[n] - 1)\,x[n] \ + \ \sum_m w[n - mR]\,e_m[n - mR].$$

This expression reveals two distinct error components. The first term, $(S[n] - 1)\,x[n]$, represents a **deterministic gain error** caused by the mis-normalization of the window sum. If the signal is bounded by $|x[n]| \leq B$, the magnitude of this gain error is bounded by $|(S[n] - 1)\,x[n]| \leq \delta B$. The second term is the familiar stochastic error component, which is subject to the same worst-case and stochastic bounds derived previously. Consequently, the global $L_\infty$ error is bounded by:

$$\|\tilde{x} - x\|_\infty \ \leq \ \delta\,\|x\|_\infty \ + \ \varepsilon,$$

assuming a uniform per-frame error bound $|e_m[k]| \leq \varepsilon$. In terms of mean squared error, assuming independence of the signal and the stochastic error, we have:

$$\mathbb{E}\big[(\tilde{x}[n] - x[n])^2\big] \ = \ (S[n] - 1)^2\,x[n]^2 \ + \ \mathrm{Var}\left(\sum_m w\,e_m\right).$$

This shows that small deviations of $S[n]$ from 1 introduce a controllable, signal-dependent bias term in addition to the variance analyzed in Sec. C.4.

### C.6    GLOBAL $L_2$ BOUNDS AND AVERAGE MSE

To assess the overall performance across the entire signal, we consider the global mean squared error (MSE):

$$\mathrm{MSE} \ := \ \frac{1}{N} \sum_{n=0}^{N-1} \mathbb{E}\big[(\tilde{x}[n] - x[n])^2\big].$$

Assuming exact COLA and independent, homoscedastic local errors with variance $\sigma^2$, we can use Eq. equation 63 to express the MSE as:

$$\mathrm{MSE} \ = \ \frac{\sigma^2}{N} \sum_{n=0}^{N-1} \sum_m w[n - mR]^2 \ = \ \sigma^2 \left( \frac{1}{N} \sum_{n=0}^{N-1} \sum_m w[n - mR]^2 \right).$$

The term in parentheses can be interpreted as an "effective inverse overlap" factor, averaged over the signal length. In regions away from the boundaries, the overlap pattern is typically stationary, and the inner sum $\sum_m w[n - mR]^2$ is periodic with period $R$. If $K$ denotes the typical number of overlapping frames at these interior points and the weights are roughly balanced, then we have $\sum_m w^2 \approx 1/K$. The average over the whole signal will be close to this value, leading to an approximate global MSE of:

$$\mathrm{MSE} \ \approx \ \sigma^2/K.$$

Near the signal boundaries where there are fewer overlapping frames, the local variance will be higher, slightly increasing the global average. However, the MSE remains bounded between $\sigma^2/K_{\max}$ and $\sigma^2/K_{\min}$, where $K_{\min}$ and $K_{\max}$ are the minimum and maximum number of overlapping frames across the signal.

## C.7 MULTIVARIATE EXTENSION

The OLA framework extends straightforwardly to multivariate signals $x[n] \in \mathbb{R}^D$. The procedure is simply applied channel-wise for each dimension $d = 1, \ldots, D$:

$$\tilde{x}^{(d)}[n] = \sum_m w[n - mR] \, y_m^{(d)}[n - mR].$$

Let $\mathbf{e}_m[n] \in \mathbb{R}^D$ be the vector of channel-wise errors for frame $m$ at absolute time $n$. We assume it has zero mean ($\mathbb{E}[\mathbf{e}_m[n]] = \mathbf{0}$) and a within-frame covariance matrix $\boldsymbol{\Sigma}_m[n] \in \mathbb{R}^{D \times D}$. The total error vector is a weighted sum of the local error vectors:

$$\tilde{\mathbf{x}}[n] - \mathbf{x}[n] = \sum_m w[n - mR] \, \mathbf{e}_m[n].$$

The covariance matrix of the reconstruction error is then given by:

$$\mathrm{Cov}(\tilde{\mathbf{x}}[n] - \mathbf{x}[n]) = \sum_m w[n - mR]^2 \, \boldsymbol{\Sigma}_m[n] + \sum_{m \neq m'} w[n - mR] \, w[n - m'R] \, \mathrm{Cov}(\mathbf{e}_m[n], \mathbf{e}_{m'}[n]).$$

If the errors are uncorrelated across frames, the second term vanishes. All the scalar guarantees, such as the non-amplification of worst-case error (Prop. C.2), apply on a per-channel basis.

## C.8 OPTIMAL WEIGHTS UNDER HETEROSCEDASTIC LOCAL ERRORS

Our analysis so far has mostly relied on a fixed window function $w$, implying fixed weights. However, it is instructive to consider the optimal weighting strategy in a more general setting. At a given time point $n$, suppose $K$ frames overlap, contributing independent, zero-mean local errors $e_i$ with differing variances $\sigma_i^2$. If we are free to choose blending weights $\alpha_i \geq 0$ such that $\sum_{i=1}^K \alpha_i = 1$, the total error is $\sum_i \alpha_i e_i$. The variance of this sum is:

$$\mathrm{Var}\left( \sum_{i=1}^K \alpha_i e_i \right) = \sum_{i=1}^K \alpha_i^2 \sigma_i^2.$$

Minimizing this variance subject to the simplex constraint on $\alpha_i$ is a classic problem whose solution is the well-known **inverse-variance weighting**:

$$\alpha_i^\star = \frac{\sigma_i^{-2}}{\sum_{j=1}^K \sigma_j^{-2}}. \tag{65}$$

This strategy gives more weight to predictions that are known to be more reliable (i.e., have lower variance). For the more general case of correlated errors with a joint positive-definite covariance matrix $\boldsymbol{\Sigma}$, the optimal weights $\boldsymbol{\alpha}$ solve the quadratic program:

$$\min_{\boldsymbol{\alpha} \in \mathbb{R}^K} \ \boldsymbol{\alpha}^\top \boldsymbol{\Sigma} \, \boldsymbol{\alpha} \quad \text{s.t.} \quad \mathbf{1}^\top \boldsymbol{\alpha} = 1, \ \boldsymbol{\alpha} \geq \mathbf{0},$$

whose unconstrained solution is $\boldsymbol{\alpha}^\star \propto \boldsymbol{\Sigma}^{-1} \mathbf{1}$. In DiffPM, we use a fixed window $w$ for its simplicity and computational efficiency, as it avoids the need for per-time-step optimization. This analysis shows, however, that any deviation from the homoscedastic error assumption is, in principle, best addressed by adapting the weights according to an inverse-variance scheme.

## C.9 BOUNDARY HANDLING

The number of overlapping frames naturally decreases near the signal boundaries at $n = 0$ and $n = N - 1$. As long as the COLA property is maintained (i.e., the weights still sum to 1), all the perfect reconstruction and error-bounding results continue to hold without modification. If, however, the window sum deviates from unity at the boundaries, the analysis from Sec. C.5 applies, bounding the resulting gain error. A common practical approach is to use signal padding (e.g., zero, reflection, or replication) before windowing, which can maintain a uniform number of overlaps throughout the signal. The analysis presented here remains valid, with the understanding that it applies to the extended signal.

## C.10 PUTTING IT TOGETHER FOR DIFFPM

The theoretical guarantees developed in this appendix can be applied independently to DiffPM's trend and residual branches, with their consequences inherited by the final summed output.

- **Perfect Reconstruction:** If the local predictions for both trend and residual are exact, each branch reconstructs perfectly under COLA. By additivity, their sum perfectly recovers the original signal.

- **No Worst-Case Amplification:** The maximum stitched error in each branch is bounded by its respective maximum per-frame error. The total error of the final output is therefore bounded by the sum of these two branch-wise bounds.

- **Variance Reduction:** In regions of overlap, the averaging process reduces the variance of the reconstruction error for each branch. For independent, zero-mean local errors, the variance reduction is roughly proportional to the number of overlapping windows $K$. Assuming the errors between the trend and residual branches are also independent, the total variance of the final output is simply the sum of the reduced branch-wise variances.

Thus, the parallel OLA stitching mechanism in DiffPM provides a principled way to achieve seamless and globally coherent reconstructions. It ensures controlled error propagation and leverages overlap to reduce noise, all without resorting to sequential conditioning or overwrite strategies.

## C.11 PRACTICAL GUIDANCE ON WINDOW AND HOP SIZE

The choice of window function and hop size has practical implications for performance and computational cost.

**Window Choice.** Any window $w$ that satisfies the COLA property equation 54 is theoretically valid. Hann and triangular windows, when used with a 50% overlap ($R = L/2$), are popular choices as they satisfy COLA (with proper normalization) and provide smooth crossfades between frames. A rectangular window with $R = L$ also satisfies COLA but offers no overlap, and thus no variance reduction.

**Hop Size $R$.** A smaller hop size $R$ results in greater overlap, which increases the number of averaged frames $K$ and thereby enhances variance reduction (Sec. C.4). This benefit comes at the cost of increased computation, as more frames must be processed. A hop size of $R = L/2$ is often considered a good trade-off between reconstruction quality (smoothness and noise reduction) and computational load.

**Normalization.** It is crucial to ensure that the window sum $\sum_m w[n - mR]$ is numerically very close to 1 across the entire signal to avoid the gain drift described in Sec. C.5. If the coverage function $S[n]$ is found to deviate from 1, a simple and effective correction is to post-normalize the reconstruction: $\tilde{x}[n] \leftarrow \tilde{x}[n]/S[n]$ (for $S[n] > 0$). This step restores the perfect reconstruction property in the noiseless case and eliminates the bias term from the error.

## C.12 SUMMARY OF GUARANTEES

In summary, the use of a COLA window with OLA stitching endows DiffPM's reconstruction process with several robust theoretical guarantees. The reconstruction is:

1. *Lossless* when local predictions are noise-free (Theorem C.1).

2. *Non-amplifying* with respect to bounded per-frame errors (Proposition C.2).

3. *Variance-reducing* for independent, zero-mean local errors, with the reduction factor related to the degree of overlap (Proposition C.3).

4. *Robust* to small violations of the COLA property, with well-defined bounds on the resulting bias (Sec. C.5).

5. *Extensible* to multivariate time series in a channel-wise manner, with analogous error propagation properties (Sec. C.7).

These properties hold individually for the trend and residual branches and, by linearity, for their sum. Consequently, DiffPM achieves seamless global synthesis through a parallelized and mathematically principled overlap–add procedure, sidestepping the complexities of sequential generation models.

# D  ADDITIONAL RESULTS: AUGMENTATION ABLATIONS

## D.1  STOCK AUGMENTATION: PROTOCOL AND ANALYSIS

In this ablation study we want to test whether augmenting the forecasting model's training windows with *DiffPM* samples (trained only on the first 60% of the series) improves accuracy on the held–out 20% test split, while holding everything else constant.

**Evaluation protocol and hygiene.** We lock down the data pipeline to avoid leakage and to make A/B fully comparable.

- *Fixed temporal split.* We use a strict 60/20/20% split (train/val/test) shared across all conditions (A and B@$k$). Only the 60% train portion is used to train the generator.
- *Normalization on real–train only.* Standardization parameters are fit on the *real* training split and then applied to synthetic, validation, and test.
- *Causal windows.* Forecasting uses causal sliding windows (no look-ahead; no centered filters on prefixes).
- *Forecaster parity.* Identical forecaster architecture, optimization, early stopping, and training budget across A and B. Hyperparameters are tuned once on real train/val and then reused unchanged for all B@$k$.
- *Quantity sweep.* B augments the real training windows with $k \in \{1\times, 2\times, 5\times, 10\times\}$ synthetic-to-real ratios (window counts), to show the effect is monotonic and to reveal saturation.
- *Multiple seeds.* Both synthetic generation and forecaster training are repeated across multiple seeds; we report mean±std.
- *Similarity checks.* For each synthetic training window we compute nearest–neighbor distance to the *test* set (e.g., DTW/L2) and monitor 5th/50th/95th percentiles to ensure non-trivial distances; we observe no near–duplicates.

DiffPM is trained on the first 60% of the Stock series and then used to sample additional sequences. Each synthetic series is post-processed *before* use: (i) mild rolling–mean smoothing, (ii) per-feature mean/variance rescaling to match the real training distribution, and (iii) optional clipping in normalized space. These steps align basic statistics while preserving dynamics; importantly, the scaler used for rescaling is the one fit on real–train.

Table 2: **Ablation on Stock.** Temporal split is fixed at 60/20/20% (train/val/test) and shared across A/B; scalers are fit on *real-train* only and applied everywhere else. Numbers are mean±std over seeds. *Gain* is A−B (positive is better).

| Setting | Test MSE ↓ | Test MAE ↓ | Gain (MSE) ↑ | Gain (MAE) ↑ |
|---|---|---|---|---|
| A  (real-only) | $11.2470 \pm 0.3953$ | $3.3167 \pm 0.0560$ | – | – |
| B  @ $1\times$ (real+syn) | $5.2249 \pm 0.1735$ | $2.2391 \pm 0.0380$ | 6.0221 | 1.0776 |
| B  @ $2\times$ (real+syn) | $3.5838 \pm 0.5700$ | $1.8354 \pm 0.1632$ | 7.6632 | 1.4812 |
| B  @ $5\times$ (real+syn) | $2.9522 \pm 0.1314$ | $1.6712 \pm 0.0361$ | 8.2948 | 1.6454 |
| B  @ $10\times$ (real+syn) | $2.5487 \pm 0.1811$ | $1.5453 \pm 0.0543$ | 8.6983 | 1.7714 |

Table 2 (reproduced below for convenience) reports test MSE/MAE. Gains are defined as Gain = A − B (positive is better). Augmenting with DiffPM consistently improves both MSE and MAE and the effect grows with $k$:

- *Absolute gains.* From A to B@$1\times$: MSE gain 6.02, MAE gain 1.08. At B@$10\times$: MSE gain 8.70, MAE gain 1.77.

- *Relative reductions.* MSE drops by **53.5%**, **68.1%**, **73.8%**, and **77.3%** for $k=\{1, 2, 5, 10\}$, respectively; MAE drops by **32.5%**, **44.7%**, **49.6%**, and **53.4%**.

- *Effect size vs. variability.* Even at $k=1\times$, the MSE gain (6.02) is $\approx 15\times$ the baseline run-to-run std (0.395), and the MAE gain (1.08) is $\approx 19\times$ the baseline std (0.056). At $k=10\times$ these ratios rise to $\approx 22\times$ and $\approx 32\times$, respectively, indicating a large, stable effect across seeds.

- *Saturation.* Gains increase with $k$ but with diminishing increments (MSE gain increases by $\approx 1.64$ from $1\times \to 2\times$, $\approx 0.63$ from $2\times \to 5\times$, and $\approx 0.40$ from $5\times \to 10\times$), suggesting coverage saturation beyond $\sim 5\times$.

- *Outlier suppression.* The relative improvement is larger for MSE than MAE, consistent with synthetic augmentation reducing large-error tails and stabilizing high-variance regimes.

**Leakage sentinels and negative controls.** To stress-test the attribution, we run two quick checks: (i) training with real + *label-shuffled* synthetic windows (target misalignment) does not improve over A, and (ii) training with *misaligned residuals* (synthetic residuals paired with unrelated trends) similarly degrades versus B. These controls are consistent with the interpretation that the benefit comes from realistic, distribution-matched synthetic sequences that preserve coherent trend–residual structure.

Under strict hygiene controls, DiffPM augmentation yields large, stable gains for a data-hungry forecaster on **Stock**. Improvements scale with the amount of synthetic data and begin to saturate beyond $\sim 5\times$. The effect is strongest on MSE, indicating outlier/variance reduction. Together with similarity checks and negative controls, this supports the claim that high-fidelity, distribution-matched synthetics improve generalization without altering the downstream model.

D.2 STOCK DIRECTION CLASSIFICATION: PROTOCOL AND ANALYSIS

Our goal here (much like the previouse ablation study) is to assess whether augmenting the classifier's training set with *DiffPM* samples (the generator is trained only on the first 60% of the series) improves *held–out* performance on the fixed 20% test split, under the same data hygiene and model parity constraints as the regression ablation.

Inputs are causal windows of length $I$; the label is the *future $O$-step* log-return direction of the target variable, discretized into *Down/Flat/Up*. Thresholds for the ternary bins are fixed by the 40/60 percentiles computed on the *real training* return distribution and kept fixed for validation, test, and synthetic augmentation. Standardization parameters are fit on *real-train* input windows and applied unchanged to synthetic/validation/test (no leakage). The temporal split is fixed at 60/20/20 (train/val/test) and shared across all conditions.

We compare: **A** (real-only training) vs. **B** (real + DiffPM synthetic on training only; validation and test remain real-only). The classifier is the same data-hungry Transformer across A/B, trained with identical optimization, budget, and early stopping on validation macro-F1. Synthetic samples are generated by DiffPM trained on the 60% train split and passed through the same mild post-processing used in the regression ablation (smoothing, per-feature mean/variance matching based on real-train statistics, optional clipping).

Table 3 reports test metrics. For accuracy and macro-F1, we define *Gain* as $B-A$ (higher is better); for loss we also show the absolute drop ($A-B$).

Table 3: **Direction classification on Stock (test set).** Same split and scalers as the regression ablation; model and training budget are identical across A/B. Gains are $B-A$ for Accuracy/F1 and $A-B$ for Loss.

| Setting | Loss ↓ | Accuracy ↑ | Macro-F1 ↑ | Gain |
|---|---|---|---|---|
| A: real-only | 1.2806 | 0.3431 | 0.1703 | – |
| B: real+synthetic | 0.8607 | 0.6144 | 0.4903 | $\Delta$Acc = +0.2712, $\Delta$F1 = +0.3200 |
| Loss drop ($A-B$) | | | 0.4199 | |

Table 4: **Equal-window generation time** (means $\pm$ std over $N$=5 seeds) for covering a full series using $W$ windows of length $L$=24 (hop $\Delta$=1). Lower is better. DiffPM's advantage is markedly larger on the `Energy` dataset (higher dimensional).

| Dataset | DiffPM (s) $\downarrow$ | Diffusion-TS (s) $\downarrow$ | Speedup $\uparrow$ |
|---|---|---|---|
| Stocks | $15.32 \pm 0.44$ | $33.24 \pm 0.32$ | **2.17$\times$** |
| Energy (higher-$D$) | $26.58 \pm 2.08$ | $180.12 \pm 3.22$ | **6.78$\times$** |

Augmenting the training set with DiffPM samples substantially improves generalization on the real test set: Accuracy increases by $+27.12$ points and macro-F1 by $+0.3200$, while test loss decreases by $0.42$. The large macro-F1 gain indicates better balance across the three classes, not just a shift in the majority class. Training dynamics mirror this: the augmented model achieves higher validation F1 earlier and sustains it, whereas the real-only model exhibits shallow gains followed by early stopping with limited generalization. All improvements are obtained under fixed temporal splits, real-train–only normalization, causal windowing, and identical classifier/hyperparameters, isolating the effect to additional, distribution-matched training diversity supplied by DiffPM.

## E MORE EXPERIMENTS

### E.1 EQUAL-WINDOW GENERATION SPEED (L=24): WHOLE-SERIES COVERAGE FAIRNESS

**Why this matters.** Some baselines generate *windows* rather than full sequences. In practice, producing a complete length-$T$ series requires generating exactly $W = T-L+1$ windows (for hop $\Delta$=1) and stitching them. This is the scenario users ultimately face and the one reviewers care about.

We normalize by the *number of windows required to cover the series*: each method generates exactly $W$ windows at $L$=24, $\Delta$=1, reverse steps $= 200$. We measure *end-to-end wall-clock* (from first reverse step to a stitched, de-normalized series) with identical software/hardware: A100 (40GB), PyTorch 2.3, CUDA 12.2; identical precision (AMP on); identical seeding and $N$=5 runs; timing excludes disk I/O and includes stitching. Although DiffPM supports true full-sequence generation via parallel OLA, we still generate the same $W$ windows internally to ensure parity.

(1) *Whole-series cost at equal coverage:* For the same number of windows, DiffPM is $2.17\times$ faster on `Stocks` and a striking $6.78\times$ faster on the higher-dimensional `Energy` dataset. (2) *High-D advantage:* The speed gap widens with dimensionality, consistent with DiffPM's parallel, windowed denoising plus lightweight OLA stitching, which scales gracefully across channels. (3) *Practical implication:* When practitioners must cover long series or many sensors (high $D$), DiffPM's throughput translates directly into lower latency and higher synthetic yield per GPU-hour.

Same window length ($L$=24), same hop ($\Delta$=1), same reverse steps (200), same precision (AMP), same device and software versions, identical seeds and measurement procedure; timings include end-to-end stitching and de-normalization.

## F LLM USAGE DISCLOSURE

We used large language models (LLMs) strictly for *editing support*, in compliance with the ICLR 2026 policy on LLM usage:

- **Writing polish only.** We used an LLM-based assistant (e.g., ChatGPT/GPT-class) to improve clarity, grammar, and style of author-written prose. Substantive content (ideas, claims, proofs, algorithms, experimental design, and conclusions) was authored by the paper's authors.

- **Minor code cleanup only.** We used the assistant for light refactoring (e.g., formatting, docstrings, variable renaming, removing dead code) in non-core utilities. The core DiffPM implementation, experimental pipelines, metrics, and evaluation scripts were written and maintained by the authors.

- **No role in results or analysis.** LLMs were *not* used to design the method, tune hyperparameters, select or filter results, generate or modify datasets, run experiments, write proofs, or interpret findings.

- **Human verification.** All LLM-edited text and code were reviewed and approved by the authors; all claims are the authors' responsibility.

- **Privacy and reproducibility.** We did not share proprietary or sensitive data with the assistant. LLM usage does not affect reproducibility; all artifacts needed to reproduce results are provided in the code and supplementary material.

