# OpenReview forum: "DiffPM: Diffusion-Based Generative Framework for Time Series Synthesis"
_ICLR.cc/2026/Conference — ICLR 2026 Conference Desk Rejected Submission_

### Official Review · Reviewer_N4Uy · 2025-10-24

**Soundness:** 2
**Presentation:** 1
**Contribution:** 2
**Rating:** 2
**Confidence:** 4

**Summary:**

The paper proposes  a diffusion-based framework for multivariate time-series synthesis. The authors aim to address challenges such as generating long sequences and maintaining both local fidelity and global consistency. The method appears to rely on a stitching strategy for composing long trajectories from shorter segments and focuses primarily on unconditional generation.

**Strengths:**

1. **Practical importance.** The paper tackles a key problem: scalable generation of long, realistic multivariate time series.
2. **Method.**
- The decomposition to residual and trend make sense, also there paper present a through analysis of the design rational and choices.
- I am not familiar with Parallel window generation and stitching method, which address  synthesis of time series generation. While it may be appear in other domains, the adaption to time series is important.

**Weaknesses:**

1. *“A formidable barrier… the difficulty of generating samples that are both locally accurate and globally consistent.”*
   The paper’s central problem statement is not substantiated. It cites neither theoretical nor empirical evidence that this issue is prevalent, nor does it provide evidence within the paper itself. Even if the problem is real, asserting it as fact without support (citations, analyses, or experiments) is scientifically unsound. The claim is reiterated again in the related work section without grounding beyond intuition, which may be misleading.


2. *“In augmentation settings, practitioners require full-length, unconditional samples — not short imputations or forecast snippets.”*
   If the central goal is long time-series generation, the paper should compare against other generative models explicitly targeting long-range synthesis [2, 4]. The reported sequence lengths are only a few hundred time steps, much shorter than those in [2]. A direct comparison on comparable long-range benchmarks would strengthen the claims.

3. The paper reports only unconditional generation results, without extending to conditional tasks (e.g., interpolation, extrapolation). This limits empirical evidence for the method’s generality and practical usefulness.

4. The paper claims the method is designed for speed; however, the empirical study supporting this claim appears only in the appendix and not in the main paper. Additionally I can't find analysis of memory consumption.

5. The evaluation omits standard discriminative metrics commonly used in time-series generative benchmarks, weakening the rigor and comparability of the results.

6. The paper does not compare against state-of-the-art diffusion-based models for time-series generation [1, 2, 3], making it difficult to assess relative performance and trade-offs. And it also omit important standard metric of discriminative evaluation.

[1] Utilizing Image Transforms and Diffusion Models for Generative Modeling
of Short and Long Time Series.

[2]  A Non-Isotropic Time Series Diffusion Model with Moving Average Transi-
tions.

[3] Generative Modeling of Regular and Irregular Time Series Data via Koop-
man VAEs.

[4] Deep Latent State Space Models for Time-Series Generation.

**Questions:**

-  In what precise sense does learning two separate denoisers confer an advantage beyond multi-scale conditioning of a single denoiser (e.g. Diffusion-TS, MR-DDPMs)?

- Does increasing overlap (smaller stride) monotonically improve metrics, or is there a quality–speed trade-off curve?

---

### Official Review · Reviewer_MiDJ · 2025-10-29

**Soundness:** 2
**Presentation:** 2
**Contribution:** 1
**Rating:** 2
**Confidence:** 5

**Summary:**

This paper proposes a decomposed diffusion model for time series generation. The core idea is to decompose the denoising process into multiple branches, to handle different signal components - trend and residual (e.g., low and high frequency) separately. The method operates on a window-based or patch-based (st-patch) principle. The authors claim this approach improves upon standard diffusion models and autoregressive methods, particularly in modeling long-range dependencies and avoiding issues like phase drift.

**Strengths:**

- The paper tackles a well-known and difficult problem in time series generation: capturing complex temporal dynamics, including both high-frequency details and long-range dependencies, which remains a challenge for many existing methods.


- The concept of decomposing the diffusion denoising process into parallel branches to specialize in different aspects of the signal is an intuitive approach.

**Weaknesses:**

This paper, in its current form, has several significant weaknesses. The experimental validation is insufficient, and key design choices are not adequately justified.

- The paper's primary weakness is the lack of any empirical comparison against established state-of-the-art (SOTA) time series generation models (e.g., other diffusion-based, GAN-based, or Transformer-based methods [1,2,3,4,5,6]). Without these baseline comparisons, it is impossible to understand the practical utility of the proposed decomposition or determine if it offers any benefits over existing work.

- The authors do not clearly differentiate their method from prior work that also employs decomposition in diffusion models. Specifically, the relationship to FiDE [1], which also decomposes the diffusion process, is not discussed. A thorough discussion and, ideally, an empirical comparison are needed to highlight the precise novel contributions of this paper over similar existing frameworks.

- The rationale for downsampling the signal is unclear, especially within a window-based generation framework. The authors state this is for a specific branch (e.g., low-frequency), but the paper does not explain why it is necessary when the generation is already localized.

- The omission of discriminative metrics is unclear and a significant gap. Such metric is standard for assessing the fidelity and practical utility of synthetic time series data and are essential for a robust comparison.

-  How the stride (referred to as "delta" in the paper) for the windowing mechanism is chosen? This parameter seems critical to how the model stitches patches together and models dependencies across windows, but no ablation study or sensitivity analysis is provided.

- The paper's central motivation is to improve long-range dependency modeling, with the authors claiming that "Autoregressive models accumulate errors over long horizons, while standard diffusion approaches can degrade long-range dependencies." However, this claim is not substantiated. The benchmarks presented appear to use very short time series (e.g., length 24 to the best of my understanding), which are insufficient to evaluate long-range dependencies. There are no experiments on longer sequences to demonstrate that the proposed method actually mitigates phase drift or correlation decay over extended horizons. This disconnect between the paper's claims and its empirical evidence is a major flaw. It is unclear if this mechanism is practical or effective in the long-range scenarios it claims to target.


[1] Galib, Asadullah Hill, Pang-Ning Tan, and Lifeng Luo. "FIDE: Frequency-Inflated Conditional Diffusion Model for Extreme-Aware Time Series Generation." NeurIPS 2024.

[2] Naiman, Ilan, et al. "Utilizing image transforms and diffusion models for generative modeling of short and long time series." NeurIPS 2024

[3] Jeon, Jinsung, et al. "GT-GAN: General purpose time series synthesis with generative adversarial networks." NeurIPS 2022

[4] Naiman, Ilan, et al. "Generative modeling of regular and irregular time series data via Koopman VAEs." ICLR 2024.

[5] Coletta, Andrea, et al. "On the constrained time-series generation problem." NeurIPS 2023.

[6] Zhicheng Chen, et al. "SDformer: Similarity-driven Discrete Transformer For Time Series Generation." NeurIPS 2024

**Questions:**

See Weaknesses.

---

### Official Review · Reviewer_yk4H · 2025-10-29

**Soundness:** 2
**Presentation:** 2
**Contribution:** 2
**Rating:** 2
**Confidence:** 4

**Summary:**

The author proposes a novel discrete framework DiffPM with disentangled diffusion structure  for time series generation task, which achieves the competitive performance of the effectiveness comparisons.

**Strengths:**

1. The paper is clear from theoretical and background aspects, less typos

2. The DIFFPM use the disentangled features, trend and seasonal parts for interoperable diffusion generation.

3. The DIFFPM shows faster inference speed, compared to the baselines.

**Weaknesses:**

1. The paper lacks the novelty, because the disentangled strategy in diffusion of time series analysis is much common, and lots of work has incorporate the disentangled strategy in the one diffusion framework rather than constructing two modules, which may cause larger memory usage.

2. The DIFFPM lack lots of popular baselines, such as , KoVAE [1], Sdformer [2], PaD-TS[3], and the generation performance show less competitive.

3. There is no ablation studies in DIFFPM, such as whether the window size may cause the different generation performance, whether the order (shuffled order) of windowed dataset may cause different results.

4. There is no description in Figure 2, which is hard to understand the process.

[1].Naiman, Ilan, et al. "Generative Modeling of Regular and Irregular Time Series Data via Koopman VAEs." The Twelfth International Conference on Learning Representations.

[2]. Chen, Zhicheng, et al. "Sdformer: Similarity-driven discrete transformer for time series generation." Advances in Neural Information Processing Systems 37 (2024): 132179-132207.

[3]. Li, Yang, et al. "Population Aware Diffusion for Time Series Generation." Proceedings of the AAAI Conference on Artificial Intelligence. Vol. 39. No. 17. 2025.

**Questions:**

Please refer to the weaknesses

---

### Official Review · Reviewer_qHcM · 2025-11-11

**Soundness:** 2
**Presentation:** 2
**Contribution:** 2
**Rating:** 2
**Confidence:** 5

**Summary:**

This paper proposes a diffusion-based time series generation framework called DiffPM. Its core idea is to first apply a moving average to decompose the time series into trend and residual components, and then train two independent diffusion models for each component. During generation, this method uses absolute position information for window sampling, generating multiple overlapping segments in parallel. Finally, these segments are concatenated by overlapping and adding them together to reconstruct the complete sequence.

**Strengths:**

1. The motivation behind this approach is clear: the multi-scale structure of time series (long-term trends and local fluctuations) is difficult to capture with a single diffusion model. Therefore, they explicitly decompose the series into trend and residual components, although this approach is common in the time series domain.

2. Experiments use cFID, correlation score, and TSTR to evaluate quality, structure fidelity, and downstream application value. The results show that DiffPM generally outperforms GAN/VAE and some diffusion baseline models, especially in preserving the multivariate correlation structure.

**Weaknesses:**

1. The decomposition relies entirely on a simple moving average with a fixed window width $K$, and there is no systematic analysis of whether this hand-crafted low/high-frequency separation is appropriate for different types of time series (e.g., strong seasonality, structural breaks, non-stationarity).

2. The main method assumes p(\tau, r)=p(\tau)p(r), i.e., the two branches are completely independent at generation time, which ignores potential strong dependencies between long-term trends and local fluctuations.

3. The ablation study is clearly insufficient: it does not thoroughly compare trend/residual decomposition vs. no decomposition, positional conditioning vs. no positional conditioning, or different window lengths; as a result, the current ablations do not fully support the claimed contribution of each design choice.

**Questions:**

Please see the above.

---

### Note · Program_Chairs · 2026-01-17
**Submission Desk Rejected by Program Chairs**

The following references in this submission do not refer to real documents and/or have major errors in bibliographic information:

 A. Paul et al. Contextual fid for time series generation. In ICLR Workshop on Time Series, 2022.
X. Ni et al. Measuring dependence in multivariate time series generation. In NeurIPS Workshop on Time Series, 2020.